# Structure-guided mutagenesis reveals a hierarchical mechanism of Parkin activation

Matthew Y. Tang[1],*, Marta Vranas[2],*, Andrea I. Krahn[1], Shayal Pundlik[1], Jean-François Trempe[2] & Edward A. Fon[1]

Parkin and PINK1 function in a common pathway to clear damaged mitochondria. Parkin exists in an auto-inhibited conformation stabilized by multiple interdomain interactions. The binding of PINK1-generated phospho-ubiquitin and the phosphorylation of the ubiquitin-like (Ubl) domain of Parkin at Ser65 release its auto-inhibition, but how and when these events take place in cells remain to be defined. Here we show that mutations that we designed to activate Parkin by releasing the Repressor Element of Parkin (REP) domain, or by disrupting the interface between the RING0:RING2 domains, can completely rescue mutations in the Parkin Ubl that are defective in mitochondrial autophagy. Using a FRET reporter assay we show that Parkin undergoes a conformational change upon phosphorylation that can be mimicked by mutating Trp403 in the REP. We propose a hierarchical model whereby pUb binding on mitochondria enables Parkin phosphorylation, which, in turn, leads to REP removal, E3 ligase activation and mitophagy.

[1] McGill Parkinson Program, Neurodegenerative Diseases Group, Department of Neurology and Neurosurgery, Montreal Neurological Institute, McGill University, Montréal, Québec, Canada H3A 2B4. [2] Groupe de Recherche Axé sur la Structure des Protéines, Department of Pharmacology and Therapeutics, McGill University, Montréal, Québec, Canada H3G 1Y6. * These authors contributed equally to this work. Correspondence and requests for materials should be addressed to J.-F.T. (email: jeanfrancois.trempe@mcgill.ca) or to E.A.F. (email: ted.fon@mcgill.ca).

Parkinson's disease (PD) is an incurable neurodegenerative disorder that is characterized by the selective and progressive loss of dopamine-secreting neurons located in the substantia nigra of the midbrain. Although the causal agent of neurodegeneration in the common sporadic form is not known, mutations in the genes that encode Parkin and PTEN-induced putative kinase 1 (PINK1) have been demonstrated to be the cause of early-onset autosomal-recessive forms of PD[1,2].

Studies initiated in *Drosophila melanogaster* have demonstrated that PINK1 acts upstream of Parkin in a common mitochondrial quality-control pathway[3,4]. PINK1 is a Ser/Thr kinase with a mitochondrial targeting sequence that is normally imported into healthy mitochondria and subsequently cleaved by proteases and degraded by the proteasome[5–9]. The kinase activity of PINK1 is directly involved in the activation and translocation of Parkin from the cytosol to the mitochondria[10]. Upon mitochondrial damage, PINK1 accumulates at the outer mitochondrial membrane where it phosphorylates ubiquitin (Ub) and the Ub-like (Ubl) domain of Parkin, at an analogous Ser65 residue[11–16]. The binding of phosphorylated Ub (pUb) to Parkin and the phosphorylation of the Ubl domain increase the E3 Ub ligase activity of Parkin, triggering a feed-forward amplification loop that results in further translocation of Parkin to the mitochondria. Once on mitochondria, Parkin activation results in the ubiquitination of several outer mitochondrial membrane targets, such as Mitofusin and Miro, which in turn induce a wide range of outcomes including proteasomal degradation, motility arrest, vesicle formation and recruitment of autophagosomes that engulf damaged mitochondria[17–21].

Parkin harbours a conserved N-terminal Ubl domain connected through a linker to four zinc-coordinating domains consisting of RING0, followed by the RBR (RING1, In-Between-RING (IBR), RING2) module common to other E3 Ub ligases. Parkin uses a RING/HECT hybrid mechanism, where its RING1 domain binds to the E2 Ub-conjugating enzyme to transfer Ub to a substrate via a thioester intermediate on the RING2 domain[22]. Structural studies revealed that Parkin, in its basal state, adopts an auto-inhibited conformation with several distinct sites of inhibition. First, the Ubl domain is bound to RING1 such that Ser65 is poorly accessible for phosphorylation by PINK1. Second, the Repressor Element of Parkin (REP) blocks the site of E2 Ub-conjugating enzyme binding on RING1. Finally, the RING0 domain binds to RING2, where it occludes the Cys431 Ub acceptor site[23–26].

The binding of pUb to Parkin has been shown to induce allosteric changes that disrupt the Ubl:RING1 interaction, which releases the Ubl domain and allows its phosphorylation by PINK1 (ref. 26). Phospho-Parkin displays a marked increase in affinity towards the E2 enzyme UbcH7, which could be emulated by mutagenesis of the REP residue Trp403 combined with deletion of the Ubl domain. Moreover, pUb also acts as a receptor for Parkin through a high-affinity interaction, effectively targeting the E3 ligase to the mitochondria[27]. This interaction is notably stabilized by electrostatic interactions between the phosphate group in pUb and His302 in Parkin, as well as hydrophobic interactions between the Ile44 patch in pUb and Ala320 in Parkin[26,28–30]. Predictably, Parkin variants that exhibit reduced binding to pUb (H302A, A320R) or that cannot be phosphorylated at Ser65 (S65A) failed to fully activate Parkin[28,29].

Yet, while it is clear that pUb binding, Ubl phosphorylation, REP release and Cys431 exposure must occur for full Parkin activation, the hierarchy of these steps and the order in which they occur on mitochondria following build-up of PINK1 remains unclear. We thus sought to investigate whether or not the decreased activity of H302A, A320R or S65A Parkin mutants can be rescued by introducing additional mutations that facilitate distinct steps in its activation. Guided by the structure, we introduced mutations that disrupt the following inhibitory interdomain contacts: (1) Ubl:RING1; (2) REP:RING1; or (3) RING0:RING2. Here we report that, in contrast to most PD-linked mutations that typically impair Parkin function, mutations in these domains result in faster translocation to depolarized mitochondria and more efficient ubiquitination of mitochondrial substrates. In addition, activating mutations that release the REP domain, or that interfere with the RING0:RING2 interface and make the active Cys431 more accessible, can rescue S65A phosphorylation-defective Parkin mutants. On the other hand, we show that binding to pre-existing pUb on mitochondria is essential for Parkin phosphorylation, and indeed deficits in Parkin recruitment and ubiquitination of outer membrane substrates by the H302A mutant cannot be rescued by any of the activating mutations. Together, these experiments provide unprecedented insight into the hierarchy and sequence of events that are involved in Parkin-mediated mitophagy, and provide a proof-of-principle that defects in Parkin-mediated mitophagy can be rescued. Considering the importance of mitochondrial quality control in PD, our findings have important implications for targeting Parkin for therapeutics in PD.

## Results

**Activating mutations enhance Parkin function**. The crystal structure of Parkin (PDB code 4K95) revealed a number of interdomain contacts that maintain auto-inhibition of its E3 Ub ligase activity (Fig. 1a). We have previously shown that mutations at the Ubl:RING1 (N273K), REP:RING1 (W403A) and RING0:RING2 (F146A) interfaces increase auto-ubiquitination activity[23,26]. If disruption of one or several of these interfaces is a rate-limiting step in Parkin activation on mitochondria, we can hypothesize that these mutations would also accelerate their translocation to depolarized mitochondria and ubiquitination of proteins at the outer mitochondrial membrane. We thus created stably expressing green fluorescent protein (GFP)-Parkin wild-type (WT) or mutant Parkin variants in U2OS cells and used time-lapse microscopy to determine the kinetics of Parkin recruitment to mitochondria after their depolarization with carbonyl cyanide m-chlorophenyl hydrazone (CCCP) (Fig. 1b,c). We have previously used the U2OS cell line to screen for modulators of Parkin recruitment since the expression of endogenous Parkin levels is undetectable relative to other cell lines[31]. Recruitment of WT Parkin to mitochondria could be detected at 30 min after CCCP treatment, whereas the W403A mutant was recruited faster than WT with a lead time of ~15 min, consistent with our previous findings[23]. The F146A mutant behaved similarly to the W403A mutant, followed by the N273K mutant with a lead time of ~5 min (Fig. 1b,c and Supplementary Fig. 1a). This suggests that releasing the auto-inhibition of the REP:RING1 or the RING0:RING2 interface similarly predisposes Parkin for mitochondrial recruitment, and does so more effectively than releasing the Ubl domain.

To exclude the possibility that the expression of Parkin mutants might indirectly affect the state of mitochondria and thus the recruitment dynamics (through changes in fusion/fission), we tested the activity of Parkin mutants in a cell-free mitochondrial ubiquitination assay adapted from a previously published protocol[32]. Briefly, mitochondria isolated from HeLa cells, which lack Parkin, were treated with CCCP to build up PINK1 at the outer mitochondrial membrane. Addition of untagged recombinant Parkin is sufficient to ubiquitinate Mfn2,

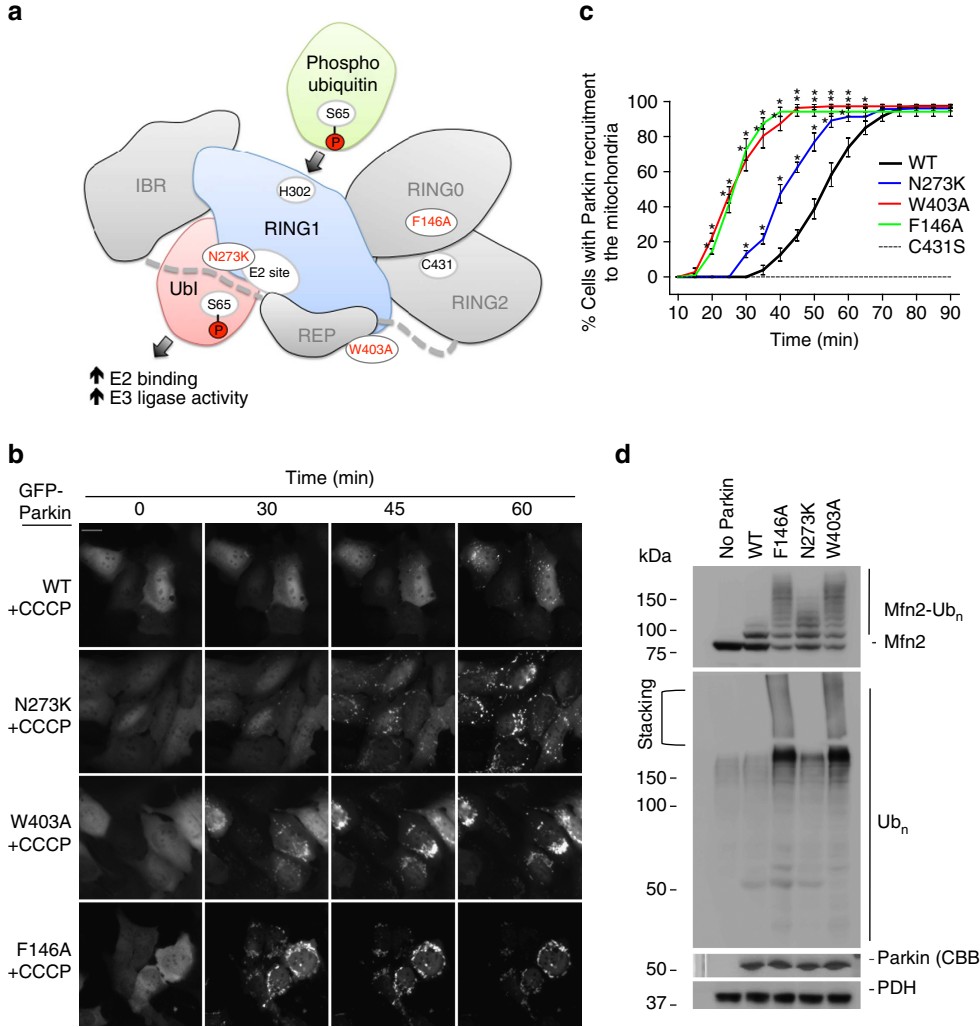

**Figure 1 | Activating Parkin mutations.** (**a**) Cartoon representation adapted from ref. 26 of the auto-inhibited structure of Parkin with activating mutations highlighted in red: F146A (RING0:RING2 interface); N273K (Ubl:RING1 interface); and W403A (REP:RING1 interface). (**b**) Time-lapse microscopy showing Parkin recruitment to mitochondria upon membrane depolarization with CCCP in U2OS cells stably expressing GFP-Parkin WT, F146A, N273K or W403A. Recruitment can be visualized by the appearance of punctate GFP fluorescence. Scale bar located at the top left of the first pictograph represents 20 μm. (**c**) Quantification of GFP-Parkin recruitment to the mitochondria. The percentage of cells showing recruitment of GFP-Parkin to mitochondria was determined every 5 min over a period of 90 min. The vertical bars represent s.e.m. from three independent experiments. *$P < 0.05$ (two-way ANOVA with Bonferroni post test). (**d**) Western blot of *in organello* ubiquitination reactions showing WT, F146A, N273K or W403A ubiquitination of Mfn2. All *in organello* ubiquitination reactions are performed with CCCP-treated mitochondria.

as observed by the formation of a high-molecular weight smear above the unmodified 80 kDa band (Fig. 1d). No activity is observed with polarized mitochondria, confirming that Parkin's activity is PINK1-dependent (Supplementary Fig. 1b). The W403A and F146A mutants showed a marked increase in polyubiquitination compared to WT, whereas the N273K mutant showed a slight increase relative to WT (Fig. 1d). Thus, using point mutations that disrupt specific inhibitory interfaces, we discovered that de-repression of all three interfaces is rate-limiting and sensitize Parkin, albeit to differing degrees.

**Phospho-Ub binding precedes Parkin phosphorylation.** The binding of pUb to Parkin acts as an allosteric modulator, dissociating the Ubl domain from RING1 and making Ser65 more accessible for phosphorylation[26,33]. The interaction between the phosphate group of pUb and Parkin occurs through the side chains of His302 and Arg305 (refs 26,28,29). As shown

previously[28], Parkin mutants that disrupt pUb binding (H302A) indeed do not translocate to depolarized mitochondria (Fig. 2a,b), nor do they ubiquitinate mitochondrial substrates in ubiquitination assays (Fig. 2c,d). In addition, disruption of the hydrophobic interaction between pUb and Parkin by the A320R mutation was found to completely block Parkin translocation[28,30] (Supplementary Fig. 2). To determine whether this effect was mediated through the inability of the Ubl domain to be released from RING1, we made Parkin double mutants that disrupt the Ubl:RING1 interface and pUb binding[26], and tested their activity in cells and *in vitro*. We found that cells expressing H302A/N273K or A320R/N273K Parkin double mutants showed significantly delayed recruitment to depolarized mitochondria compared to WT (Fig. 2a,b and Supplementary Fig. 2). The release of the Ubl domain is therefore insufficient to rescue mutant Parkin variants defective in pUb binding. Binding of pUb to Parkin was also proposed to help release the REP and affect the RING0:RING2 interface[28]. Therefore, we next asked

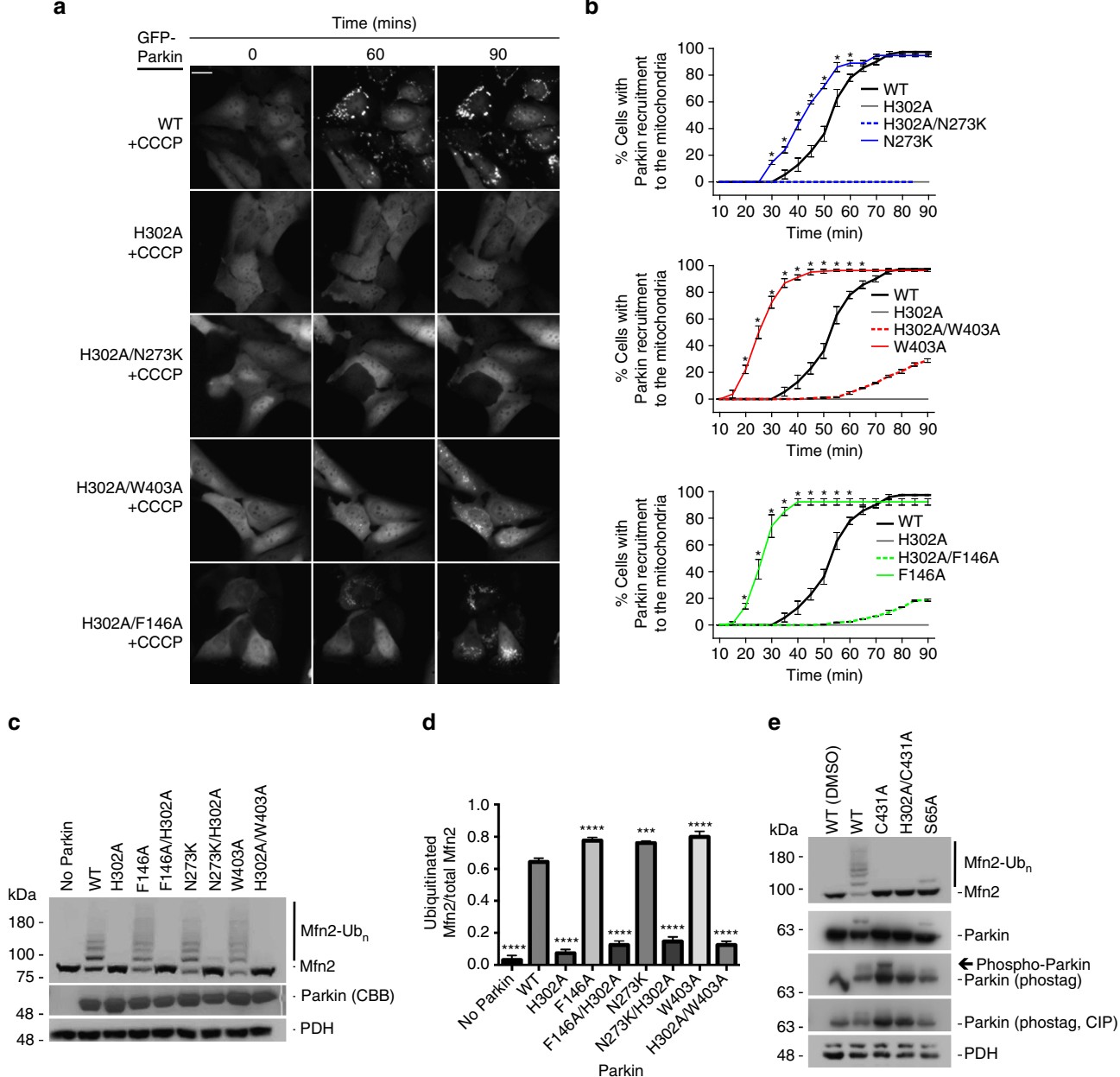

**Figure 2 | Effects of Parkin mutants that disrupt pUb binding on Parkin recruitment and Mfn2 ubiquitination.** (**a**) Time-lapse imaging of Parkin recruitment to mitochondria upon treatment with CCCP in U2OS cells stably expressing WT, H302A, H302A/N273K, H302A/F146A or H302A/W403A Parkin. Recruitment can be visualized by the appearance of punctate GFP fluorescence. Scale bar located at the top left of the first pictograph represents 20 μm. (**b**) Quantification of GFP-Parkin recruitment to the mitochondria. The percentage of cells showing recruitment of GFP-Parkin to mitochondria was determined every 5 min over a period of 90 min. The vertical bars represent s.e.m. from three independent experiments. $*P < 0.05$ (two-way ANOVA with Bonferroni post test). Data from WT and hyperactive mutants from experiments carried out in parallel from Fig. 1 are shown for comparison in each graph. (**c**) Western blot of *in organello* ubiquitination reactions showing WT, H302A, H302A/N273K, H302A/F146A or H302A/W403A ubiquitination of Mfn2. All *in organello* ubiquitination reactions are performed with CCCP-treated mitochondria. (**d**) Quantification of Mfn2 ubiquitination signals from three independent experiments. The vertical bars represent s.e.m. $***P < 0.001$; $****P < 0.0001$ (one-way ANOVA with Dunnett's test). (**e**) Western blot of *in organello* ubiquitination reactions correlating defects in pUb binding to phosphorylation of Parkin on mitochondria. *In organello* reactions were either loaded directly on gel or treated with Calf Intestinal Alkaline Phosphatase (CIP, NEB) for 30 min at 37 °C and then loaded on a gel. Arrow indicates phosphorylated Parkin. All *in organello* ubiquitination reactions are performed with CCCP-treated mitochondria unless stated differently.

whether activating mutations that release the REP or disrupt the RING0:RING2 interface could rescue deficits in pUb binding by making Parkin double mutants (H302A/W403A, H302A/F146A, A320R/W403A or A320R/F146A). Neither the W403A- nor F146A-activating mutations were able to rescue deficits in Parkin recruitment and ubiquitination (Fig. 2a–d and Supplementary

Fig. 2), although low levels of mitochondrial recruitment could be observed in the H302A/W403A and H302A/F146A mutants at late time points. This confirms that the role of pUb as a Parkin receptor is primordial.

We have previously shown that the H302A mutant does not completely abrogate binding of pUb, but reduces its $K_d$ to 1.3 μM

compared to 22 nM for the Parkin-Δ1-140 (Ubl- and linker-deleted)[26]. We thus hypothesized that the low levels of mitochondrial recruitment of Parkin observed at late time points could be explained by residual low-affinity binding of H302A Parkin to pUb. We therefore reasoned that increasing the concentration of the H302A mutant in *in organello* ubiquitination assay would rescue its Mfn2 ubiquitination defect. We indeed find that increasing the concentration of H302A to values matching those of the calculated $K_d$ increases Mfn2 ubiquitination ratios. Interestingly, 1–10 µM of H302A/N273K and H302A/W403A mutants show increased activity compared to that of H302A alone (Supplementary Fig. 3a,b). This is consistent with pUb acting primarily as a high-affinity receptor for Parkin, effectively enabling its phosphorylation and activation in proximity to PINK1. To test this idea, we used Phos-tag SDS–polyacrylamide gel electrophoresis (SDS–PAGE) gels to assess phosphorylation of different Parkin mutants upon incubation at low concentration (100 nM) with mitochondria containing PINK1. To avoid the confounding effect of Parkin-generated Ub chains leading to further recruitment and pUb generation, and thus probe the initial phosphorylation step, we mutated the active site Cys431 to alanine to abolish its ligase activity. We found that Parkin C431A becomes phosphorylated, but that the H302A/C431A double mutant does not (Fig. 2e). This suggests that binding to pre-existing pUb on mitochondria is a prerequisite to downstream Parkin phosphorylation. Consistent with this idea, we were able to detect pUb in our mitochondrial samples in the absence of Parkin (Supplementary Fig. 3c–e), as observed previously[14]. Together, these data are consistent with a model, whereby binding of Parkin to low-abundance mitochondrial-tethered pUb is the first step, which in turn triggers Parkin phosphorylation by PINK1 and leads to a feed-forward amplification loop promoting efficient recruitment to the mitochondria and ubiquitination of outer mitochondrial membrane proteins.

**Ubl release cannot rescue defects in Parkin phosphorylation.**
Phosphorylation of Parkin Ubl Ser65 has been shown to activate Parkin[11–13]. Parkin phosphorylation not only weakens the Ubl:RING1 interactions, but also enhances Parkin's affinity towards pUb and its E3 Ub ligase activity[26]. To better understand the role of Ubl Ser65 phosphorylation in Parkin activation, we specifically addressed the effect of Ubl release by creating a S65A/N273K double mutant and compared its activity against the single mutants. As expected, the S65A mutant showed a delay in Parkin recruitment compared to WT when treated with CCCP (Fig. 3a,b). The S65A/N273K mutant was not able to rescue the S65A in Parkin recruitment assays (Fig. 3a,b), nor was it able to rescue the defects in ubiquitination of Mfn2 in our *in organello* ubiquitination assays (Fig. 3c,d). Therefore, releasing the Ubl domain is insufficient to overcome a defect in Ubl Ser65 phosphorylation or a defect in pUb binding, suggesting that the role of Ubl Ser65 phosphorylation is not simply to release the Ubl from RING1 and enhance its affinity for pUb.

**REP and RING0 mutants rescue defects in Ubl phosphorylation.**
We have previously shown that a mutation of the REP anchor residue Trp403 enhances the affinity of Parkin for UbcH7 similarly to the effect of Parkin phosphorylation at Ser65 of the Ubl domain. This suggested that phosphorylation of Ser65 activates Parkin by releasing the REP domain[26]. Thus, we hypothesized that activating mutations might rescue Parkin defective for Ubl phosphorylation. We therefore generated Parkin double mutants at Ser65 combined with mutations that disrupted the REP domain (S65A/W403A) or the RING0:RING2

interface (S65A/F146A). The S65A/W403A and S65A/F146A double mutants were both recruited to depolarized mitochondria faster than the S65A mutant and even slightly faster than WT (Fig. 3a,b and Supplementary Video 1). In addition, we found that the S65A/W403A and S65A/F146A double mutants were able to ubiquitinate Mfn2 *in organello* (Fig. 3c,d). Thus, the removal of the REP domain or the disruption of the RING0:RING2 interface that expose the catalytic Cys431 during Parkin activation is sufficient to rescue defects in Ubl Ser65 phosphorylation. Moreover, we found that the W403A or F146A mutants were also able to rescue the recruitment activity of ΔUbl-Parkin, but the N273K mutant could not rescue as expected (Fig. 4a,b). Moreover, the effect was independent of the linker between Ubl and RING0, since both Δ1–140 and Δ1–76 Parkin could be rescued by the W403A mutation *in organello* (Fig. 4c,d and Supplementary Fig. 4). This suggests that the main role of the phosphorylated Ubl is to dislodge the REP from RING1, which favours E2 binding. In support of this model, we find that phospho-Parkin, as well as the W403A mutant in the WT, S65A or ΔUbl contexts, discharges UbcH7∼Ub faster than the corresponding construct with intact Trp403 (Supplementary Fig. 5a,b). Moreover, those same mutants show enhanced reactivity towards haemagglutinin (HA)-tagged Ub vinyl-sulfone, which selectively reacts with the reactive Cys431 thiol group (Supplementary Fig. 5c,d). These results confirm that disruption of the REP:RING1 and RING0:RING2 interface mimics Ubl phosphorylation and shifts Parkin towards an active conformation that facilitates the concerted thioester transfer from the E2∼Ub conjugate to Cys431.

**REP and RING0 mutants enhance mitophagy.** In light of the findings that Parkin mutants that disrupt the REP:RING1 or RING0:RING2 interfaces have increased rates of mitochondrial recruitment, ubiquitination of mitochondrial substrates and can rescue deficits in Parkin phosphorylation, we sought to assess the effects of activating mutations on mitochondrial turnover by autophagy. To monitor and quantify mitophagy dynamically, we used a sensitive fluorescence-activated cell sorting (FACS)-based mitophagy assay consisting of U2OS cells stably expressing inducible mt-Keima, a pH-sensitive fluorescent protein that is targeted to mitochondria and exhibits a large shift in its emission wavelength upon engulfment in the acidic compartment of lysosomes[34]. Lysosomal-positive mt-Keima was detected upon mitochondrial depolarization with CCCP or a combination of antimycin A and oligomycin (Supplementary Fig. 6a,b) within 2 h (Supplementary Fig. 6c,d). In our assay, we found that both the W403A and F146A mutants, but not the N273K mutant, showed higher levels of mitophagy compared to WT at 4 h (Fig. 5a,b and Supplementary Fig. 7). This is consistent with our recruitment and ubiquitination results that showed greater activity in these mutants. Thus, disruption of the REP:RING1 or RING0:RING2 interfaces, but not the release of the Ubl domain (N273K), increases mitophagy.

Next, we tested whether our de-repression mutants could rescue deficits in mitophagy caused by loss of pUb binding. As expected, the H302A mutant defective in pUb binding was completely deficient in mitophagy compared to WT at 4 h (Fig. 6a–c). We find that both the W403A and F146A double mutants could rescue the H302A mutations in the mt-Keima mitophagy assay, but not mutations that release the Ubl domain (N273K) (Fig. 6a–c). This result is in qualitative agreement with our recruitment results, which showed partial rescue for the two former double mutants (Fig. 2a,b), and also implies that fast recruitment kinetics is not essential to achieve maximal autophagy rate. It also implies that at least in the

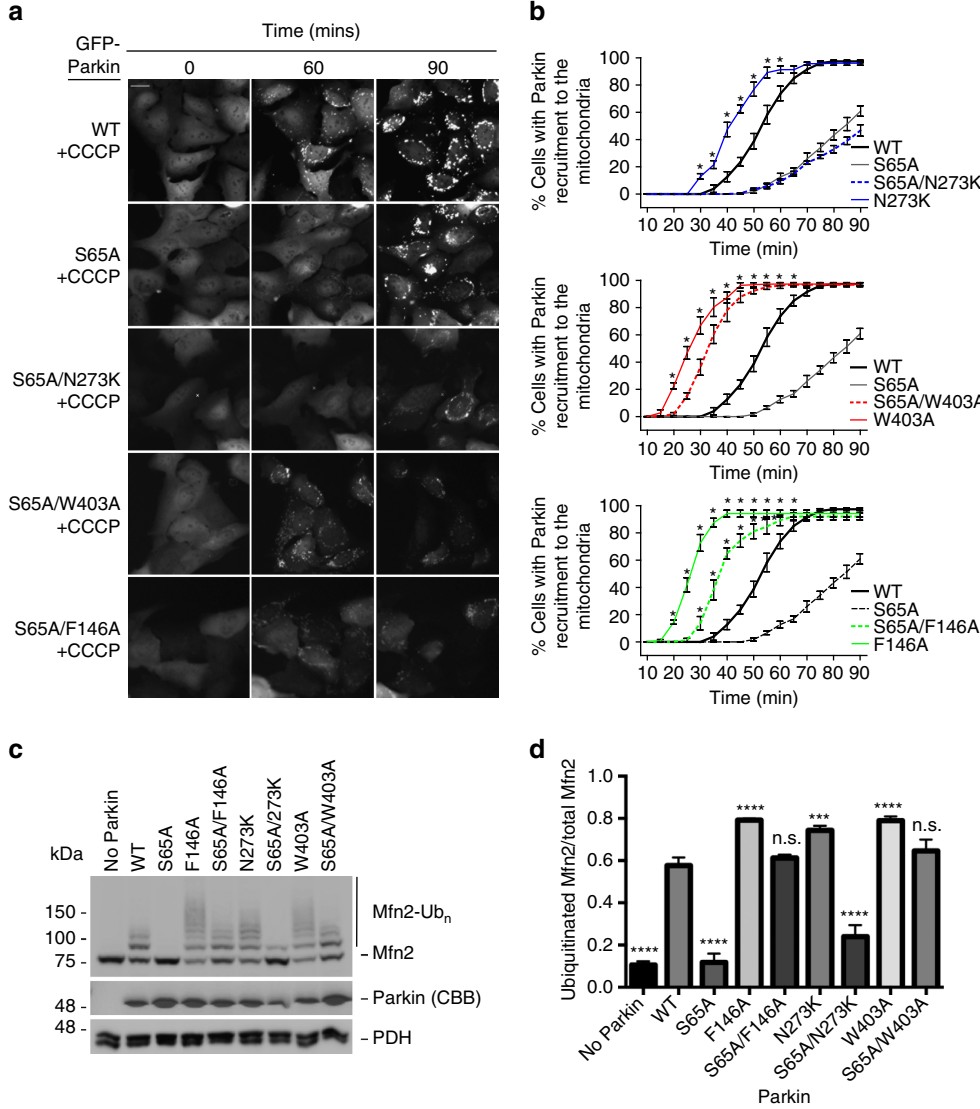

**Figure 3 | Rescuing the S65A Parkin variant recruitment to the mitochondria using activating mutations.** (**a**) Time-lapse microscopy showing recruitment of GFP-Parkin variants to mitochondria upon treatment with 20 μM CCCP. Recruitment can be visualized by the appearance of punctate GFP fluorescence. U2OS cells stably expressing S65A/N273K, S65A/W403A or S65A/F146A Parkin double mutants were compared to WT, S65A or N273K Parkin. Scale bar located at the top left of the first pictograph represents 20 μm. (**b**) Quantification of GFP-Parkin recruitment to the mitochondria for cells expressing S65A/N273K, S65A/W403A or S65A/F146A double mutations. Data from WT and hyperactive mutants from experiments carried out in parallel from Fig. 1 are shown for comparison in each graph. The percentage of cells showing recruitment of GFP-Parkin to mitochondria was determined every 5 min over a period of 90 min. The vertical bars represent s.e.m. from three independent experiments. For statistical analysis, a two-way ANOVA with Bonferroni post test was performed, *$P < 0.05$. (**c**) Western blot of *in organello* ubiquitination assays for the rescue of the phospho-dead S65A mutant. (**d**) Quantification of Mfn2 ubiquitination signals from three independent experiments. The vertical bars represent s.e.m. ***$P < 0.001$; ****$P < 0.0001$; n.s., nonsignificant (one-way ANOVA with Dunnett's test). All *in organello* ubiquitination reactions are performed with CCCP-treated mitochondria unless stated differently.

*in vivo* conditions of our cellular mitophagy assay it is possible to overcome the reduced pUb-binding affinity of the H302A Parkin mutant by disrupting the inhibitory REP:RING1 or RING0:RING2 interfaces. We then tested whether activating mutations could rescue mitophagy caused by the loss of Ubl phosphorylation. We found that the S65A mutant had more mitophagy than the H302A or A320R mutant, but less than WT (Fig. 6b–d and Supplementary Fig. 8a,b). Interestingly, both the W403A and F146A mutants could rescue the S65A (Fig. 6b–d) and ΔUbl mutants (Supplementary Fig. 9a,b) in our mt-Keima mitophagy assays, which is again consistent with our recruitment and ubiquitination assays. Overall, this implies that disruption of either the REP:RING1 or RING0:RING2

interface can rescue mitophagy deficits in Parkin mutants impaired in pUb binding or phosphorylation.

**Phosphorylation and REP mutant induce similar conformations.** The rescue of the Parkin Ubl phospho-dead mutation S65A by the REP mutation W403A is consistent with the previous observation that Ser65 phosphorylation enables E2 binding by disrupting the REP:RING1 interaction[26,29,35]. We therefore sought to determine whether the W403A mutation and Ser65 phosphorylation lead to similar conformational changes using Förster resonance energy transfer (FRET) microscopy. We made a series of FRET reporter constructs with the donor Cerulean

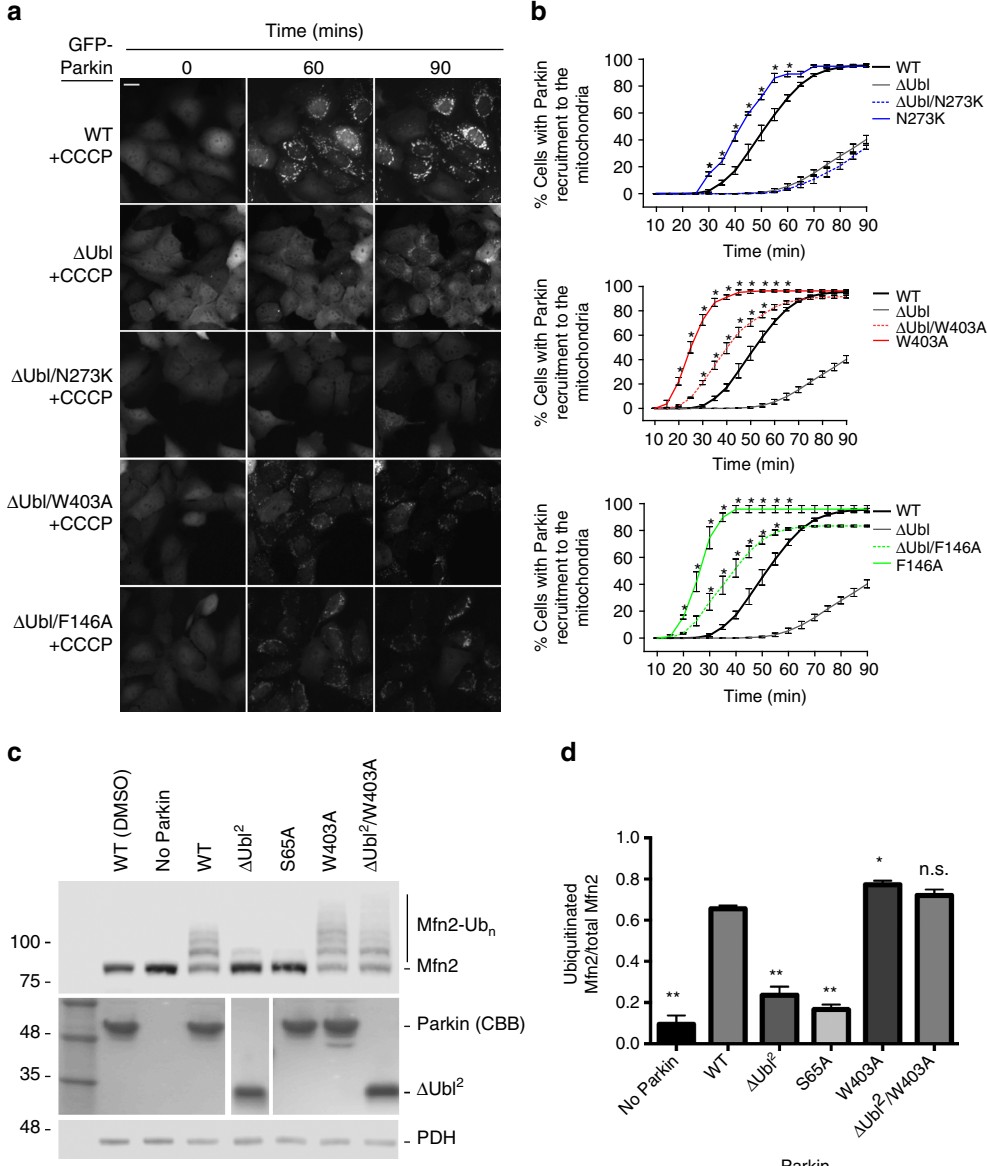

**Figure 4 | Deletion of the Ubl impairs recruitment and mitochondrial substrate ubiquitination.** (**a**) Time-lapse microscopy of Parkin recruitment to mitochondria upon treatment with CCCP in U2OS cells stably expressing WT, ΔUbl (deletion of residues 1–76), ΔUbl/F146A or ΔUbl/N273K Parkin double mutants. Recruitment can be visualized by the appearance of punctate GFP fluorescence. Scale bar located at the top left of the first pictograph represents 20 μm. (**b**) Quantification of GFP-Parkin recruitment to the mitochondria for Parkin double mutant-expressing cells. The percentage of cells showing recruitment of GFP-Parkin to mitochondria was determined every 5 min over a period of 90 min. The vertical bars represent s.e.m. from three independent experiments. *$P < 0.05$ (two-way ANOVA with Bonferroni post test). Data from WT and hyperactive mutants from experiments performed in parallel from Fig. 1 are shown for comparison in each graph. (**c**) Western blot of *in organello* ubiquitination assays for the rescue of ΔUbl (ref. 2; deletion of residues 1–140). (**d**) Quantification of Mfn2 ubiquitination signals from three independent experiments. The vertical bars represent s.e.m. *$P < 0.05$, **$P < 0.01$; n.s., nonsignificant (one-way ANOVA with Dunnett's test). All *in organello* ubiquitination reactions are performed with CCCP-treated mitochondria unless stated differently.

fluorescent protein (CFP) at the N terminus of Parkin, and the acceptor yellow fluorescent protein (YFP) Venus inserted into linkers connecting different domains in Parkin to detect conformational changes (Supplementary Fig. 10a,b), using the crystal structure of full-length rat Parkin as a canvas[23]. These constructs are expected to show a FRET signal due to the proximity of CFP and YFP within Parkin[36]. HeLa cells were transfected with our FRET reporter constructs, and FRET microscopy measurements were performed (Fig. 7). While the fluorescent proteins may have restrained motion that limit distance-based interpretation of the FRET efficiency measurements, the basal values measured for our reporter

constructs are in good qualitative agreement with the Parkin structure (PDB code 4K95), with the FRET-380 construct (YFP inserted after residue 380 within CFP-Parkin) yielding the shortest distance and highest FRET efficiency (Fig. 7). Thus, our FRET measurements show that Parkin expressed in mammalian cells adopts a conformation that is generally consistent with its crystal structure.

To ensure that the inserted fluorescent proteins did not interfere with the function of the protein, all constructs were tested in a mitochondrial recruitment assay following addition of CCCP as well as the FACS-based mitophagy assay. The recruitment kinetics of Parkin with YFP inserted at positions

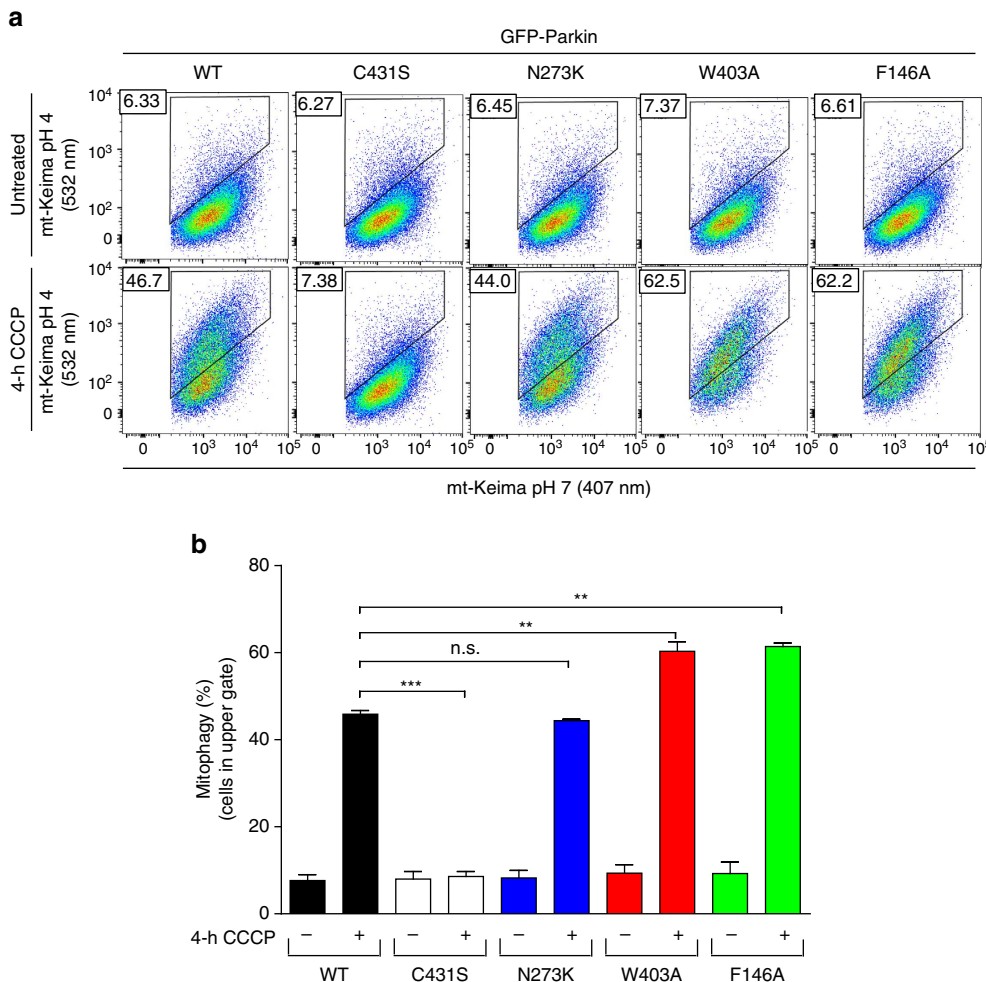

**Figure 5 | Activating mutations in Parkin show higher levels of mitophagy.** (**a**) Mitophagy was examined using a FACS-based analysis of mitochondrially targeted mKeima (mt-mKeima). Representative FACS data of mt-mKeima expressing WT or hyperactive Parkin mutants untreated or treated with CCCP for 4 h from two experiments. (**b**) Quantification of average percentage of mitophagy. $**P < 0.001$; $***P < 0.0001$; n.s., nonsignificant (one-way ANOVA with Tukey's post test).

81 (after Ubl), 129 (Ubl-RING0 linker) and 380 (IBR-REP linker) were similar to the kinetics of N terminally tagged Parkin (Supplementary Fig. 11), and were able to induce mitophagy (Supplementary Fig. 12a,b). Moreover, purified recombinant FRET-81, FRET-129 and FRET-380 proteins are stable and susceptible to activation by phosphorylation by PINK1 as confirmed by an *in vitro* auto-ubiquitination assay (Supplementary Fig. 12c). In contrast, insertion at positions 140 (before RING0) delayed recruitment and insertion of YFP at positions 356 (within loop in IBR) and 466 (C terminus) strongly impaired recruitment (Supplementary Fig. 11), and thus these constructs were not analysed further.

Co-transfection of CFP-Parkin and YFP-Parkin displayed no FRET, implying that the basal FRET observed in our dual CFP–YFP constructs is unlikely to arise from intermolecular interactions (Fig. 7). To determine whether conformational changes resulting from the disruption of auto-inhibitory interfaces in Parkin could be measured using our FRET reporter constructs, we introduced the N273K-, F146A- or W403A-activating mutations in FRET-81 and FRET-380—the two FRET reporter construct with the highest basal FRET efficiency. All three activating mutations in the FRET-380 reporter showed a reduction in basal FRET efficiency (Fig. 8a–c). In contrast, no reduction in FRET efficiency was observed in the FRET-81 reporter (Fig. 8b,c). In fact, the N237K mutations in the FRET-81

reporter showed an increase in the FRET efficiency, which may arise from a change in the motion and anisotropy of the fluorescent proteins following Ubl release from RING1. The FRET-380 reporter with its high basal FRET efficiency is therefore more suitable for monitoring conformational changes associated with Parkin activation. Disruption of both the Ubl:RING1 interaction (N273K) and the RING1:REP interface (W403A) in the FRET-380 reporter showed the greatest reduction in FRET efficiency, which is consistent with a model where the REP and Ubl domains change conformation upon activation. The disruption of the RING0:RING2 interface (F146A) also showed a significant, yet small decrease in FRET. To exclude the possibility that the decrease in FRET in FRET-380 was the result of E2-Ub-activating enzyme binding to Parkin, we introduced a mutation that was previously shown to inhibit E2 binding (T240R) and created a double T240R/W403A mutant. It was found that the T240R/W403A double mutant had a similar decrease in basal FRET efficiency observed with the W403A mutation (Supplementary Fig. 13).

To determine whether the phosphorylation of Ser65 is capable of inducing a change in FRET efficiency in our FRET-380 reporter, we created a phosphomimetic mutation S65E and measured basal FRET efficiency. Interestingly, we found that S65E had a reduction in FRET efficiency in the FRET-380 reporter that was not observed in the S65A or WT reporter

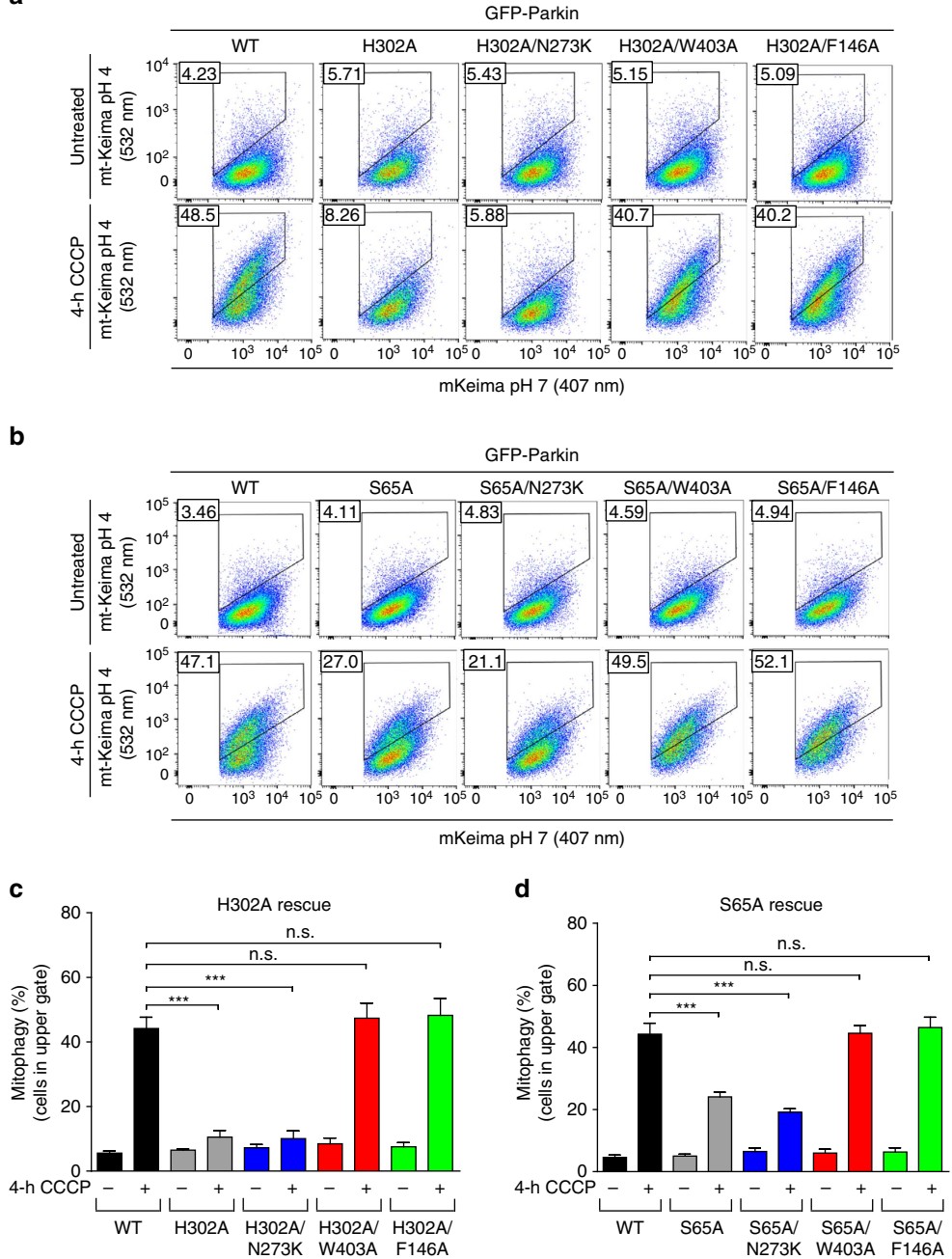

**Figure 6 | Activating mutations can rescue defects in mitophagy.** (**a**,**b**) Representative FACS data of mt-mKeima U2OS cells expressing WT or Parkin mutants untreated or treated with 20 μM CCCP for 4 h. (**c**,**d**) Quantification of average percentage of mitophagy from three independent experiments. Vertical bars represent s.e.m.. For statistical analysis, a one-way ANOVA with Tukey's post test was performed. ***$P < 0.0001$; n.s., non significant.

(Fig. 8d,e). These observations are consistent with the idea that phosphorylation of Parkin at Ser65 induces a conformational change that is similar to mutations that disrupt the RING1:REP domain (W403A). Taken together, these experiments strongly suggest that Parkin adopts a different conformation during Parkin activation that results in a displacement of the Ubl and the REP, which can be monitored using our FRET-380 reporter.

## Discussion
Mutations in *Parkin* linked to PD are inherited in an autosomal-recessive manner and are typically associated with a loss of function characterized by a decrease in its E3 Ub ligase activity.

Here we provide evidence that disruption of Parkin auto-inhibitory interfaces that release the Ubl, REP or expose the active site Cys431 can be used to enhance its activity both *in vitro* and in cells. In particular, we find that the designer REP mutation W403A is the most effective at enhancing Parkin activity as well as rescuing defects in pUb binding and Ser65 phosphorylation. Conversely, disruption of the Ubl:RING1 interface with the N273K mutation could not rescue defects in pUb binding or Ser65 phosphorylation, even though the N273K single mutant is still functional and recruits to the mitochondria and ubiquitinates mitochondrial substrates faster than WT. The N273K mutant indeed binds better than WT Parkin to pUb[26], which could explain the faster activation kinetics. Yet, while

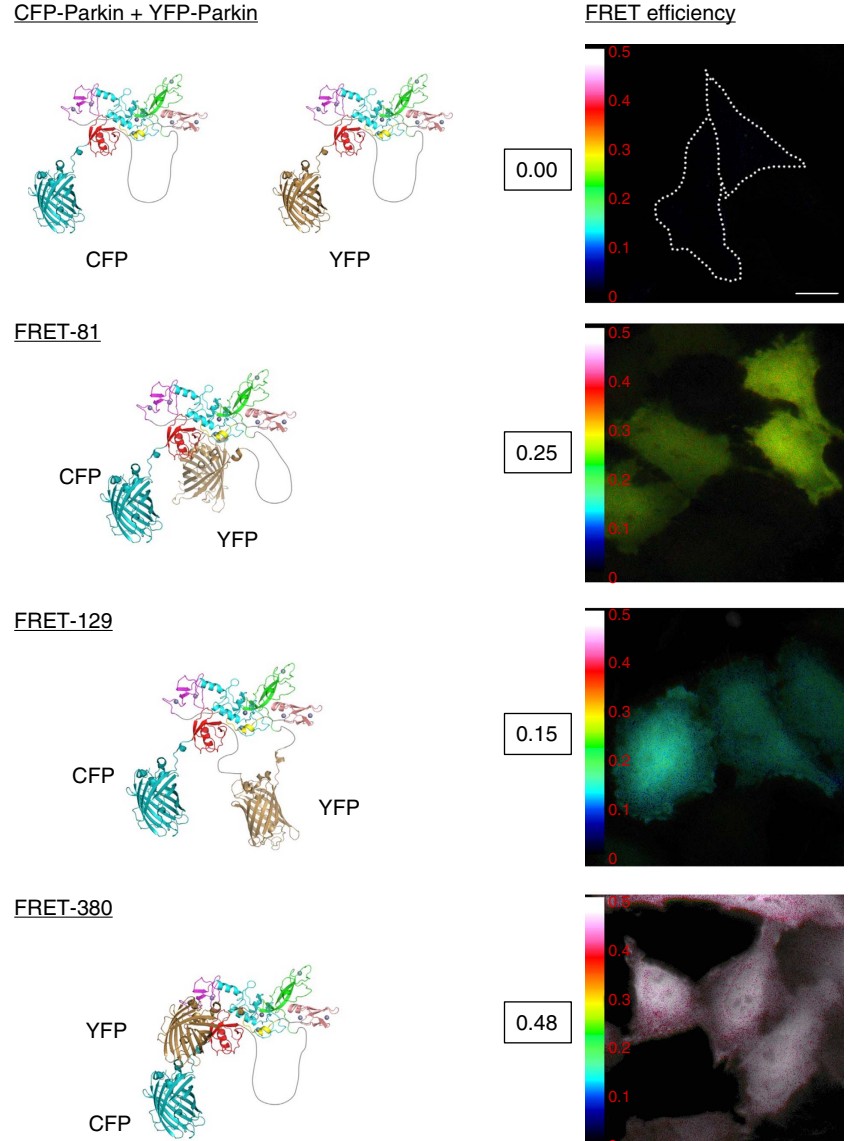

**Figure 7 | Characterization of Parkin FRET reporters.** Intramolecular FRET measurements in Parkin. HeLa cells were transfected with FRET constructs consisting of Parkin tagged with Cerulean (CFP) at its N terminus and Venus (YFP) at different positions. Molecular models of tagged Parkin constructs were derived from the crystal structure of Parkin and GFP. FRET Xia levels were measured and shown using a colour-coded FRET map to display the measured FRET ratio (warm colours represent higher levels of FRET and cooler colours represent lower levels of FRET). The average FRET efficiency is indicated next to each FRET map with a scale ranging from 0.00 to 0.50. The FRET signal correlates qualitatively with the distance between CFP and YFP in the molecular models. Scale bar, 20 μm.

release of the Ubl is certainly a required step for Parkin activation, it is insufficient to fully activate Parkin *per se*. Consistent with this idea is the observation that ΔUbl–Parkin shows a delay in recruitment to depolarized mitochondria[37] (Fig. 4), in spite of the Ubl deletion actually increasing its affinity for pUb[26]. Phosphorylation of the Ubl at Ser65 therefore plays a positive role in Parkin activation by stabilizing a conformation where the REP adopts a different configuration that allows E2 binding, which can be mimicked by the W403A mutation[26].

Previous studies have proposed that binding of Parkin to pUb is required to trigger the feed-forward amplification loop that promotes Parkin's E3 ligase activity and leads to mitochondrial recruitment and ubiquitination of outer mitochondrial membrane proteins[14,27,35]. However, others have suggested that Parkin phosphorylation precedes pUb generation based on the observation that Parkin phosphorylation occurs prior to the

build-up of significant amount of pUb[38]. Here we provide evidence that the binding of Parkin to pre-existing pUb on mitochondria is required for its phosphorylation (Fig. 2e). Binding to pUb allosterically promotes PINK1-mediated phosphorylation of the Parkin Ubl domain by disrupting the Ubl:RING1 interaction, thereby making Ser65 accessible to the kinase[26]. Consistent with the role of pUb as a mitochondrial receptor, we find that the ubiquitination of Mfn2 by WT or H302A Parkin in *in organello* ubiquitination assays follow concentration-dependence patterns that correlate with their respective affinities for pUb (Supplementary Fig. 3a,b). This observation could explain why the W403A and F146A mutations are able to partially rescue defects in mitochondrial recruitment and mitophagy in H302A mutant in cells, where the GFP-Parkin concentration may be as high as 1–10 μM (that is, equal or above its $K_d$ of 1.3 μM). However, no rescue was

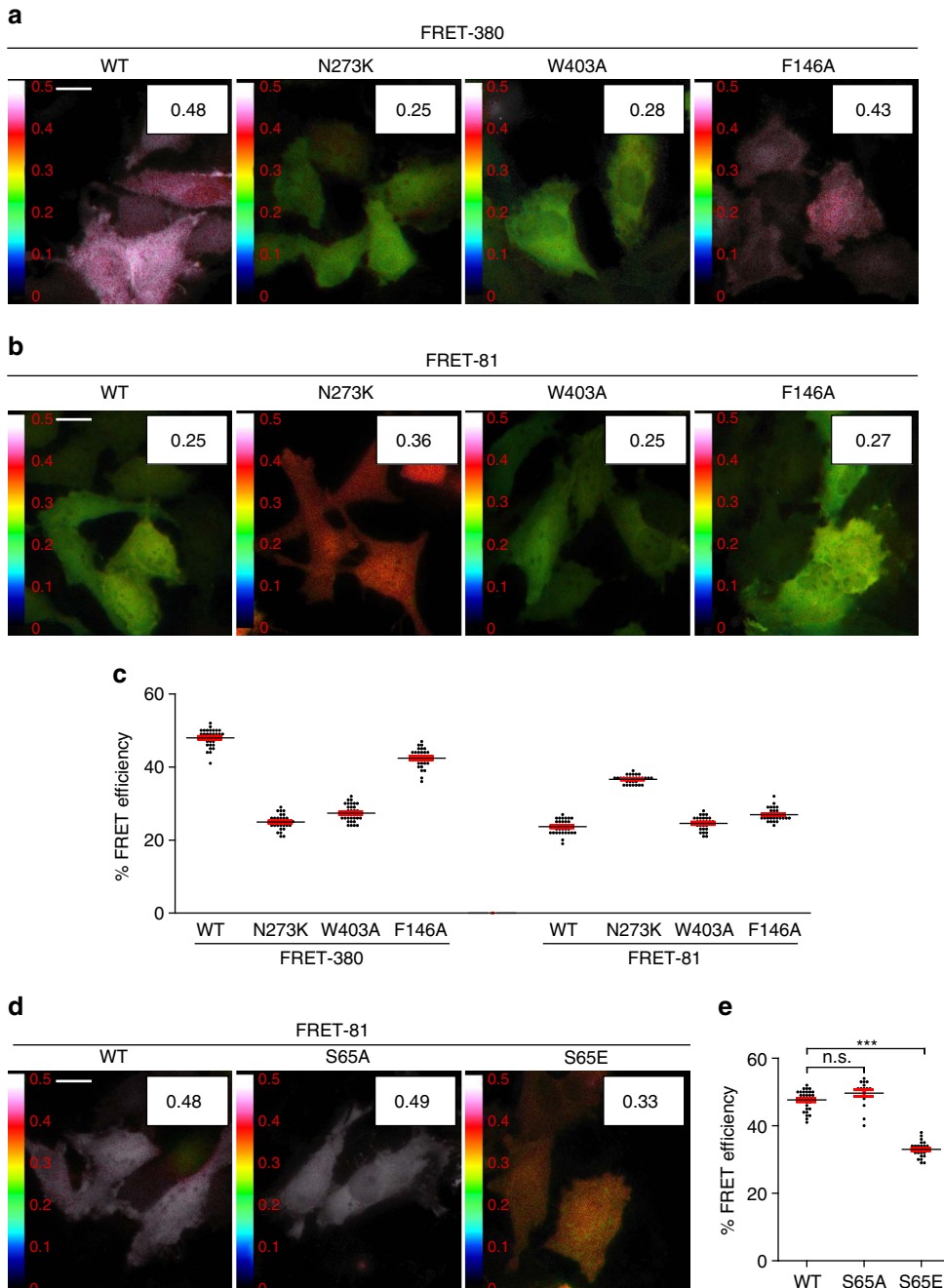

**Figure 8 | Single-point mutations in FRET reporters.** (**a**,**b**) FRET images and average FRET efficiency of WT and activating point mutations N273K, W403A and F146A in the FRET-380 and FRET-81 reporters. (**c**) Quantification of FRET efficiency measurements from **a**,**b**. Each data point represents the FRET efficiency measurement from one cell. (**d**) FRET efficiency measurements and (**e**) quantification of the S65E phosphomimetic in the FRET-380 reporter. Vertical bars represent s.e.m. for three independent experiments. ***$P < 0.0001$; n.s., not significant (one-way ANOVA with Tukey's post test). Scale bars, 20 μm.

observed in cell-free ubiquitination assays where we maintain Parkin at 100 nM, which may be a more physiological concentration. Indeed, levels of overexpressed Parkin are much higher than endogenous concentrations[39,40].

In our rescue experiments, we found that both F146A and W403A mutations were able to rescue the S65A mutation in mitochondrial ubiquitination, recruitment and early mitophagy. Yet, the FRET measurements revealed a significant difference between W403A and F146A. Because the YFP protein was inserted into flexible regions of Parkin where weak or no electron density was observed, we cannot predict with certainty how

the fluorescent proteins will be positioned and how restricted its motion will be. The N- and C termini of the fluorescent protein are separated by 20–25 Å, and this gap may affect the structure and motion of the reporter constructs as well, although we have shown that these insertions clearly do not affect the function of Parkin (Supplementary Figs 11 and 12). Since the fluorescent protein may not tumble isotropically, the Förster equation does not hold and cannot be used to measure distances. Nonetheless, FRET value changes can be compared to determine whether conformational changes induced by different mutations are of a similar nature. While the W403A

mutation induces a conformational change that is highly similar to that of the phosphomimetic S65E mutation in the FRET-380 reporter (Fig. 8d,e), the F146A mutation showed only a modest decrease in FRET efficiency (Fig. 8a–c). Another piece of evidence that suggests that the F146A mutation induces a distinct conformational change is that it considerably reduced solubility in the purified recombinant Parkin protein, whereas the W403A mutant behaved similarly to WT. For that reason, we were unable to perform *in vitro* UbcH7~Ub discharging and Ub vinyl-sulphone (UbVS) reactivity assays with the F146A-containing mutants, since the protein would precipitate during the assay. Overall, our results suggest that disruption of the RING0:RING2 interface induces a conformational change distinct from disrupting the REP:RING1 interaction, but nonetheless favours the transition towards the active conformation.

The enhancement of Parkin's activity by mutagenesis raises interesting prospects for PD therapy. Indeed, the levels of soluble Parkin in the brain decrease with age[40] and PD patients show higher levels of insoluble inactive Parkin compared to non-affected individuals[41]. The W403A mutation is able to increase the intrinsic activity of Parkin and bypasses the need for Ser65 phosphorylation. Critically, this mutant still requires PINK1 for mitochondrial autophagy[23]. The W403A mutation thus sensitizes Parkin for translocation to damaged mitochondria only because the localization of Parkin is mediated by PINK1-generated pUb. The FRET tools we have developed, in particular the FRET-380 construct, will facilitate the identification and characterization of compounds that can recapitulate the effects of Ser65 phosphorylation and Trp403 mutagenesis, with the potential for Parkin therapeutics in PD.

## Methods

**Cell culture.** Human osteosarcoma U2OS cells stably expressing GFP-Parkin was a gift from Dr Robert Screaton (Sunnybrook Research Institute). All other GFP-Parkin mutants were created by transfecting plasmid DNA using GeneJuice (EMD Millipore), followed by selection with G418 for 2 weeks and finally were sorted using flow cytometry. *Homo sapiens* full-length Parkin was cloned in pEGFP-C2 (Clontech) and mutated using PCR mutagenesis according to the manufacturer's protocol (Agilent Technologies). Constructs were verified by sequencing and immunoblotting using rabbit anti-GFP (Invitrogen). Cells were maintained in DMEM supplemented with 10% fetal bovine serum (FBS), 4 mM L-glutamine and 0.1% Penicillin/Streptomycin, and were maintained in a 37 °C incubator with 5% $CO_2$.

**Constructs for *H. sapiens* Parkin proteins.** To make the FRET reporter constructs, Parkin was amplified from enhanced GFP (eGFP)-Parkin (previously described in ref. 23) using PCR with primers containing HindIII and BamH1. The product of this amplification was then cut with HindIII and BamH1, and inserted into pCDNA3 cut with the same enzymes to make pcDNA-Parkin. The sequence of Venus from the Venus-N1 construct was inserted at various sites within Parkin by PCR and fragment exchange (DNA Express Inc., Montreal). The resulting construct was amplified by PCR using primers containing BglII and HindIII. The product of this amplification was then cut with BglII and HindIII and ligated to the mCerulean-C1 vector cut with the same enzymes. The mVenus-N1 and mCerulean-C1 constructs were gifts from Dr Steven Vogel (Addgene plasmid #27793 and #27796). No linkers were inserted between Parkin and the fluorescent proteins, except for the FRET-466. To make the FRET-466 construct, Parkin was amplified from eGFP–Parkin using PCR with primers containing BglII and HindIII with a mutation introduced to remove the stop codon of Parkin. The product of this amplification was cut with BglII and HindIII and inserted into Cerulean-C1 cut with the same enzymes. The resulting plasmid was amplified in dam-/dcm-competent *Escherichia coli* cells (New England BioLabs), cut with HindIII and XbaI and ligated with the Venus fragment resulting from cutting Venus-N1 with HindIII and XbaI. The complete sequence for each FRET reporter construct is found in Supplementary Fig. 14.

**Cloning and production of purified recombinant proteins.** *Hs*Parkin FRET reporter sequences were cloned from mCerulean-C1 vectors into pGEX-6P-1 by Gibson Assembly. Codon optimization of *Rattus norvegicus* Parkin and *Dendroctonus ponderosae* PINK1 (123–570) for *E. coli* expression was performed by gene synthesis (DNA Express Inc.). The genes were cloned into pGEX-6P-1 using BamH1 and Xho1 restriction enzymes. Single-point mutations were

introduced in codon-optimized *Rn*Parkin using PCR site-directed mutagenesis. Protein expression in BL21 (DE3) *E. coli* cells was induced with 25 µM isopropy-β-D-thiogalactoside (IPTG) for the different Parkin variants and 100 µM IPTG for *Dp*PINK1. Protein purification was performed as previously described[26]. Briefly, proteins were purified by glutathione-Sepharose agarose affinity and eluted with 20 mM glutathione, followed by 3C cleavage and size-exclusion chromatography on Superdex 200 16/60 (GE Healthcare) in 50 mM Tris-HCl pH 7.5, 120 mM NaCl and 1 mM dithiothreitol (DTT). His6-tag UbcH7 was produced in BL21 (DE3) *E. coli* cells transformed with pET28a-LIC-UbcH7 (ref. 23). Protein was purified by Ni-NTA (Qiagen) and eluted with 300 mM Imidazole, followed by gel filtration on Superdex 75 16/60 (GE Healthcare) in 10 mM HEPES pH 7.5, 50 mM NaCl and 1 mM DTT. Protein concentrations were determined with ultraviolet absorption at 280 nm using the theoretical extinction coefficients.

**Auto-ubiquitination assays.** Ubiquitination assays were performed for 45 min at 37 °C in 50 mM Tris-HCl pH 7.5, 120 mM NaCl, 1 mM DTT, 4 mM ATP, 50 µM Ub, 10 mM $MgCl_2$, 50 nM E1, 2 µM UbcH7 and 2.5 µM *Rn*Parkin with or without 2 µg of glutathione *S*-transferase (GST)-*Dp*PINK1. Reactions were stopped with the addition of SDS–PAGE sample buffer containing 100 mM DTT and analysed using gel electrophoresis and Coomassie staining.

**In vitro phosphorylation of Parkin by *Dp*PINK1.** Parkin phosphorylation was performed with 3.5 µM GST-*Dp*PINK1 and 20 µM of *Rn*Parkin (in 50 mM Tris-HCl, 120 mM NaCl and 1 mM DTT), 5 mM $MgCl_2$ and 1 mM ATP in a reaction volume of 20 µl at 30 °C for 60 min.

**Charging and discharging assay of UbcH7.** UbcH7 (10 µg) was charged with N-terminal fluorescein-labelled Ub (10 µg FluoUb) using 0.02 µg $His_6$-E1 (Boston Biochem Inc.), 0.5 mM ATP and 10 mM $MgCl_2$ in a 100 µl reaction (in 50 mM Tris/HCl pH 7.4, 120 mM NaCl and 1 mM TCEP). After 60 min incubation at 30 °C, 0.8 µg of UbcH7~FluoUb was mixed with 1.2 µg of WT or mutant *Rn*Parkin (in 50 mM Tris/HCl pH 7.4, 120 mM NaCl and 1 mM TCEP). Mixtures were incubated for 20 min at 30 °C for discharging. Reactions were stopped with the addition of SDS–PAGE sample buffer containing 12 mM TCEP, resolved on SDS–PAGE and analysed using the Typhoon fluorescent scanner (GE) and Coomassie staining. Fluorescence strength of UbcH7~FluoUb was measured for quantification using ImageJ.

**Modification of Parkin by UbVS.** N-terminal Ha tag UbVS (1 µg) was added to WT or mutant *Rn*Parkin (3 µg) in 50 mM Tris/HCl pH 7.4, 120 mM NaCl and 1 mM TCEP in a 50 µl reaction. After a 10-min incubation at 37 °C, reactions were stopped with 3× sample buffer with 100 mM DTT and analysed by western blotting. After transfer to nitrocellulose, membranes were blocked with 5% milk in PBS-T (0.1% Tween 20) and incubated with rabbit anti-Parkin (1:2,000, Ab15954 AbCam) or rabbit anti-HA (1:2,000 monoclonal antibody (mAb) C29F4, Cell Signaling), diluted in PBS-T with 3% BSA. Membranes were washed with PBS-T and incubated with horseradish peroxidase (HRP)-coupled goat anti-rabbit IgG antibodies (1:10,000, Cell Signaling). Detection was performed with Clarity Lightning ECL (Bio-Rad) and images acquired with a ImageQuant LAS 500 (GE Healthcare). Quantification of HA and Parkin was performed by quantifying western blot signals using the ImageJ software.

**Mitochondrial GFP-Parkin recruitment time-lapse microscopy.** Human osteosarcoma U2OS cells ($7.5 \times 10^4$) stably expressing the various GFP–Parkin constructs were seeded on a 35-mm Glass Bottom Microwell Dish (MatTek Corporation). After 24 h, cells were transferred on a heated stage maintained at 37 °C and at 5% $CO_2$ using a Zeiss temperature controller and cell perfusion system (Zeiss). To visualize mitochondria, cells were transduced with Cell Light mitochondrial-RFP (Invitrogen) as per the manufacturer's protocol. Cells were treated with CCCP (Sigma), a proton ionophore, at a final concentration of 20 µM. Microscopy was performed on a Zeiss AxioObserver.Z1 inverted fluorescent microscope. Fully automated multidimensional acquisition was controlled using the Zen Pro software (Zeiss). Images were acquired using a ×20 objective (Plan-Apochromat 0.8) with a side-mounted AxiocamMRm camera. GFP or red fluorescent protein was excited using the Zeiss XBO75 Xenon illumination system and detected using the appropriate filters (62HE ExGFP in combination with a Hyper GFP reflector or CY3, respectively). Fixed exposure times were as follows: brightfield phase contrast 10 ms; GFP 100 ms; and RFP 180 ms. Images were taken at 5 min intervals for a total of 90 min and compiled into movie files using the Zen Pro software. For the live-cell analysis of GFP–Parkin recruitment on mitochondria, 350 cells were examined over three separate experiments to ascertain the time required for GFP-Parkin to be recruited on mito-RFP-positive mitochondria. Parkin recruitment upon membrane depolarization is visualized by the appearance of punctate GFP fluorescence superposed on mitochondria RFP fluorescence. Quantification of GFP-Parkin recruitment to mitochondria is facilitated by calculating the percentage of cells showing recruitment of GFP-Parkin on mitochondria at 5 min intervals over a period of 90 min. The fluorescence intensity of the GFP-positive puncta was not taken into

consideration for Parkin recruitment analysis. For statistical analysis, a two-way analysis of variance (ANOVA) with Bonferroni post test was performed, *$P < 0.05$.

**Mitochondria isolation and *in organello* ubiquitination assay.** HeLa cells treated with 10 μM CCCP or dimethylsulphoxide (DMSO) for 3 h were suspended in mitochondrial isolation buffer (20 mM HEPES/KOH (pH 7.4), 220 mM Mannitol, 10 mM KAc and 70 mM sucrose) on ice. Cells were disrupted by nitrogen cavitation, and cell homogenates were centrifuged at 600$g$ for 5 min at 4 °C to obtain a post-nuclear supernatant. Cytosolic fractions were collected by two further centrifugation steps for 10 min at 4 °C, the first at 10,000$g$ and the second at 12,000$g$. Mitochondria pellets were suspended in mitochondria isolation buffer to a concentration of 2 mg ml$^{-1}$ and stored at −80 °C until further use. Forty micrograms of CCCP- or DMSO-treated mitochondria were supplemented with an ubiquitination reaction mix (20 nM Ub-activating enzyme 1 (E1), 100 nM of Ub-conjugating enzyme 2 (E2), 5 μM Ub, 1 mM ATP, 5 mM MgCl$_2$ and 50 μM TCEP in mitochondria isolation buffer) and 100 nM of recombinant *Rn*Parkin. After a 30-min incubation at 37 °C, reactions were stopped with 3 × sample buffer with 100 mM DTT and analysed by western blotting. Reactions were loaded on 8% Tris-glycine gels or on 8% Tris-glycine gels containing 20 μM Phos-tag and 40 μM MnCl$_2$ for separation of phosphorylated proteins (Fig. 2e). Proteins were transferred to nitrocellulose and stained with Ponceau. Membranes were blocked with 5% milk in PBS-T (0.1% Tween 20) and incubated with rabbit anti-mitofusin2 (1:2,000, mAb D2D10, Cell Signaling), mouse anti-Parkin (1:40,000, mAb Prk8, Cell Signaling), rabbit anti-PINK1 (1:2,000, mAb D8G3, Cell Signaling), rabbit anti-phospho Ub S65 (1:2,000, ABS1513 Millipore), rabbit anti-PDH (1:2,000, mAb C54G1, Cell Signaling) or rabbit anti-VDAC (1:5,000, mAb D73D12, Cell Signaling) diluted in PBS-T with 3% BSA. Membranes were washed with PBS-T and incubated with HRP-coupled goat anti-mouse or anti-rabbit IgG antibodies (1:10,000, Cell Signaling). Detection was performed with Clarity Lightning ECL (Bio-Rad) and images acquired with a ImageQuant LAS 500 (GE Healthcare). Uncropped images corresponding to Figs 1d, 2c,e, 3c and 4c are found in Supplementary Fig. 15.

Quantification of mitofusin 2 ubiquitination was performed by quantifying western blot signals of *in organello* ubiquitination assays using ImageJ (v1.48, NIH). For statistical analysis, a one-way ANOVA followed by a Dunnet's test was performed using Graph Pad Prism 6 (La Jolla, CA). All statistical analyses were run on data obtained from three independent experiments. Findings were considered significant as follows: *$P < 0.05$; **$P < 0.01$; ***$P < 0.001$; ****$P < 0.0001$; n.s. nonsignificant.

**Mass spectrometry.** Mitochondria were isolated from HeLa cells as described above. Mitochondrial proteins were extracted by methanol–chloroform precipitation[42]. Briefly, four volumes of methanol, one volume of chloroform and three volumes of water were added to one volume of mitochondria suspension, mixing the solution after each addition. After 5 min centrifugation at 13,000$g$, the upper aqueous phase was discarded. Three volumes of methanol were added and the sample centrifuged again after mixing. Pelleted proteins were dried by evaporation and resuspended in denaturing buffer (6 M urea, 1 mM EDTA, 50 mM TEAB pH 8.5). Cysteine residues were then reduced with TCEP (2 mM pH 7.0) and alkylated with iodoacetamide (10 mM solution in H$_2$O freshly prepared, Sigma). Protein samples were diluted to 1 M urea and digested with 1:100 trypsin (Sigma) overnight at 37 °C. Digested peptides were purified using ZipTip C18 pipettes (Millipore). Peptides were diluted in loading buffer (2% acetonitrile, 0.05% trifluoroacetic acid), and 2 μg of peptides were captured and eluted from an Acclaim PepMap100 C18 column (75 μm × 15 cm) with a 2 h gradient of acetonitrile in 0.1% formic acid at 300 nl min$^{-1}$. The eluted peptides were analysed with an Impact II Q-TOF spectrometer equipped with a Captive Spray nanoelectrospray source (Bruker). Data were acquired using data-dependent automatic tandem mass spectrometry (auto-MS/MS) with a range 150–2,200 $m/z$ range, a fixed cycle time of 3 s, a dynamic exclusion of 1 min, $m/z$-dependent isolation window (1.5–5 Th) and collision energy 25–75 eV (ref. 43). MS/MS data were analysed with MASCOT using a standard search procedure against the human proteome, and extracted ion chromatograms were produced using the Data Analysis software from Bruker.

**Mitophagy assay.** Mitophagy was examined using a FACS-based analysis of mitochondrially targeted mKeima (a gift from A. Miyawaki, Laboratory for Cell Function and Dynamics, Brain Science Institute, RIKEN, Japan). U2OS cells stably expressing an ecdysone-inducible mt-Keima were induced with 10 μM ponasterone A, transfected with GFP-Parkin for 18 h and treated with 20 μM CCCP for 4 h. For flow cytometry analysis, cells were trypsinized, washed and resuspended in PBS prior to their analysis on a LSR Fortessa (BD Bioscience) equipped with 405 and 561 nm lasers and 610/20 filters (Department of Microbiology and Immunology Flow Cytometry Facility, McGill University). Measurement of lysosomal mitochondrially targeted mKeima was made using a dual-excitation ratiometric pH measurement where pH 7 was detected through the excitation at 405 nm and pH 4 at 561 nm. For each sample, 100,000 events were collected and single, GFP-Parkin-positive cells were subsequently gated for mt-Keima. Data were analysed using FlowJo v10.1 (Tree Star). For statistical analysis, a one-way ANOVA

with Tukey's post test was performed on data from two independent experiments. *$P < 0.05$; **$P < 0.001$; ***$P < 0.0001$; n.s., nonsignificant.

**FRET and image quantification.** FRET reporter constructs consisting of Parkin tagged with Cerulean (CFP) at its N terminus and Venus (YFP) at different positions were transfected into HeLa cells for 24 h prior to FRET measurements. Images were acquired using a Zeiss AxioObserver.Z1 fluorescent microscope using an X-cite Series 120 illumination system with a side-mounted QuantEM:512SC EMCCD camera, a × 63 oil objective (Plan-Apochrom 1.4), and were detected using appropriate FRET filters (Zeiss filter set 47 HE Cyan, 46 HE Yellow and 48 CFP–YFP-FRET). Images of cells expressing CFP alone and YFP alone were taken as control measures for bleed-through subtraction. Each cell was imaged in the CFP, YFP and FRET configurations with constant exposure times for all experiments. FRET efficiency was quantified using the Zeiss AxioVision software, with CFP assigned as channel 1, YFP as channel 2 and FRET as channel 3. Regions of interests in cells where both CFP and YFP fluorescences were detected were selected for FRET efficiency measurements. The FRET Xia formula was used to subtract spectral bleed-through from the CFP and YFP channels[36]. The FRET ratio ranged from 0 to 0.5, which was expressed as a corresponding colour-coded FRET ratio map with warm colours represented higher levels of FRET and cooler colours represented lower levels of FRET. Samples were randomized prior to image acquisition and quantification. Region of interest from at least 30 cells for each experiment from three independent experiments were analysed for FRET efficiency. For statistical analysis, a one-way ANOVA with Tukey's post test was performed.

**Data availability.** The authors declare that data supporting the findings of this study are available within the paper and its Supplementary Information files.

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

## Acknowledgements

We thank members from the Trempe and Fon labs, as well as Kalle Gehring for useful discussion and comments. We would like to thank Carole Farah for technical assistance with the FRET microscopy. The McGill SPR-MS Facility thanks the Canada Foundation for Innovation (CFI) for infrastructure support. We acknowledge support from Parkinson Society Canada (New Investigator Award to J.-F.T. and Basic Science Postdoctoral Fellowship to M.Y.T.), the Michael J. Fox Foundation (J.-F.T and E.A.F.), the FRQ-S Groupe de Recherche Axé sur la Structure des Protéines (J.-F.T.), the Canada Research Chair Program (J.-F.T.), the Parkinson's Disease Foundation grant number PDF-FBS-1548 (M.Y.T.) and the Canadian Institutes of Health Research grant MOP-62714 (E.A.F.).

## Author contributions

M.Y.T. performed all experiments in cells (microscopy, FACS and FRET), M.V. performed all protein purification, *in vitro* and *in organello* ubiquitination assays, A.I.K. assisted with FACS and S.P. assisted with FRET analysis. M.V. and J.-F.T. performed the mass spectrometry experiment. M.Y.T., M.V., J.-F.T. and E.A.F. participated in the design of experiments, data analysis and preparation of the manuscript.

## Additional information

**Competing financial interests:** The authors declare no competing financial interests.

