## [Peer Review File · Nature Communications]

Reviewers' Comments:

Reviewer #1 (Remarks to the Author)

The Manuscript by M.Y. Tang et al. entitled 'Structure-guided mutagenesis reveals a hierarchical mechanism of parkin activation' describes the activation mechanism of the protein Parkin which has been described as a key regulator of Parkinson's disease. A series of point mutations were selected based on a mentioned protein crystal structure and are further investigated in a series of assays for their effect on Parkin's ability to undergo mitochondrial translocation, ubiquitination, and supposed conformational changes investigated by FRET.

Overall, the manuscript has good potential. However, in its current state the manuscript suffers from some weaknesses, particular in the FRET section, that make it difficult to follow the author's conclusions.

Mitochondrial translocation assay:

What is the cut-off used by the authors to judge a cell as positive for the appearance of punctate GFP-parkin? If one looks at figure 1 panel b and compares the 60min picture for wt or N273K my rough estimate would be that wt is about 20-30% signal of N273K. However, if one compares this to the quantified time response data in panel c, the difference at time point 60min is roughly 25% between wt (75-80%) and N273K (100%).

Is intensity considered at all, or would any weak spot be taken as positive punctate pattern?

This discrepancy needs to be clarified and the assay criteria need to be better defined, since the assay is used throughout the manuscript and its results represent a major part of figures 1-4.

FRET-probe:

This section suffers from quite a number of weaknesses.

First: Probe names

On page 11 the authors state:

'The recruitment kinetics of Parkin with YFP inserted at positions 81 (after Ubl), 129 (Ubl-RING0 linker) and 380 (IBR-REP linker) were similar to the kinetics of N-terminally tagged Parkin. In contrast, insertion at positions 140 (before RING0) delayed recruitment and insertion of YFP at positions 356 (within loop in IBR) and 465 (C-terminus) strongly impaired recruitment (Supplementary Fig. 5bc), and thus these constructs were not analyzed further.'

465 (C-terminus) is irritating since the probe can be found as 466 (c-terminus) in table b supplementary figure 5. Please match nomenclature....

Position 140 shows delayed recruitment and 356 and 465 show strongly impaired recruitment. However, in table b supplementary figure 5 the recruitment after 1h CCCP is less for 140, much less for 356 and again less for 466.

What are the author's criteria to convert delayed recruitment and strongly impaired recruitment to less or much less?

Second: The description of the probe design is extremely vague and it is currently impossible for a skilled person to reproduce any of their findings.

On page 16 in the methods section the authors state:

The sequence of Venus from the Venus-N1 construct was inserted at various sites within Parkin by

PCR and fragment exchange (DNA Express Inc.). The resulting construct was amplified by PCR using primers containing BglII and HindIII.

This is the shortest and most insufficient description of a FRET probe I have seen in a long time! The authors need to state the use of linkers or should state that no linkers were used. If no linker were used, how does the distance in N-term and C-term of Venus influence the parkin protein? The insertion of Venus will split parkin in two parts and eventually insert a distance of up to 2.2 nm between both fragments (2.2 nm by simply measuring the distance in GFP pdb code 1emb, Glu6 to Ile229 given as 22.38 Å). This needs to be taken into account and discussed!

Figure 7 FRET-129:

The structural illustration of YFP is strongly misleading since it is connected to parkin via a single amino acid link. If it is inserted into parkin, it has to have two connections to parkin.

The distance interpretation in all FRET Probes has to be revised by the authors with great care. An inserted Venus is restrained in its degrees of freedom and hence, unlikely to freely rotate as a fluorescent protein could do if it was tethered at the N- or C-term of a protein. Therefore, any distance interpretation is a wild guess since the assumption of $K=2/3$ as value for the Förster equation does not hold true. It could be anything between 0 and 4. Without anisotropy measurements, it is impossible to draw any conclusion with respect to distances! Therefore, the interpretation of the results needs to be revised with greater care!

Minor Points:

Page 5:

`The crystal structure of Parkin revealed a number of inter-domain contacts that maintain auto-inhibition of its E3 ubiquitin ligase activity (Fig. 1a).`

Please provide the reference for the mentioned crystal structure, since it is very irritating for none specialists to go to figure 1 and discover a cartoon rather than a structure as expected after this sentence....

Figure 3. The labeling of the individual figure panels is from a-d and does not match the labeling in the figure legend a-b, e-f. Obviously or likely, the figure has been modified from a previous version. Please correct labeling in the figure legend to a-d.

Last but not least: I would be interested to know if any of the described mutation occur in patients suffering from Parkinson's disease. Is there anything known?

Reviewer #2 (Remarks to the Author)

There is a pressing need therapeutically and biologically to understand how the Parkin E3 ligase that is mutated in Parkinson's disease is activated. The Fon group previously solved the crystal structure of Parkin that revealed Parkin to be in a complex auto inhibited conformation at several interfaces. Together with other groups they have demonstrated that phosphorylation of Parkin at its Ubl domain and of ubiquitin together lead to rearrangement of the Parkin structure into an active conformation with concomitant activation of Parkin E3 ligase activity. In this paper they address mechanistically the step-wise order by which each of these autoinhibitory interfaces are released to lead to activation. Overall they provide evidence that phospho-ubiquitin binding to Parkin is the initial step in activation and that Parkin phosphorylation is not simply leading to a release of the Ubl domain but plays an active role in causing release of the REP element that binds

RING1 and the RING0/RING2 auto inhibition towards activation. The analysis undertaken is elegant and the results extremely clear.

Whilst several of their conclusions have already been made in previous studies published last year and this year by other groups, their approach is distinct due to the comprehensive nature of the analysis undertaken to study all the auto inhibitory interfaces together and using both in vitro and cellular assays of Parkin recruitment, activity and mitophagy.

Overall I have only a few technical suggestions to strengthen the conclusion of the paper:

1. Whilst they show that the Ubl/RING1 disruptor mutant, N273K, is more active than WT in Parkin recruitment and activity (Fig1b-d), this mutant is totally unable to rescue defects in Parkin Ser65 phosphorylation. This is not completely expected given the increased activity and could there be a specific interaction here between the N273K and S65A residues that blocks activity. It would be important to test different Ubl domain mutants that have previously shown to affect the Ubl/RING1 interface e.g. the R33Q or A46P to confirm this result.

2. The effects of the activating mutations on mitophagy at 4h show no effect with the N237K mutant despite this also being more active than WT. It would be good to see a time course of CCCP-induced mitophagy to compare the mutants with WT more comprehensively.

3. The translocation assay to check FRET constructs is not sufficient to assess impact on function. To verify that the insertion of YFP into residues 81, 129 and 380 is silent, it would be important to look at other downstream readouts of Parkin activity in cells upon CCCP and also express them in E.coli to check in vitro activity.

Minor points:

1. On page 5, results section, the times stated for WT parkin recruitment of 30mins post CCCP does not reflect the data shown in Fig1 and supp Fig 1. Also true for N237K lead time mutant.

2. Bottom of page 7 - should read Fig 2e and not 3e

3. The labelling of Fig 6c x-axis should be switched from R305A to H302A

Reviewer #3 (Remarks to the Author)

Parkin and PINK1 are the gene products of familial Parkinson's disease, and are essential for the clearance of damaged mitochondria. PINK1 accumulated on the damaged mitochondria phosphorylates both ubiquitin and the Ubl domain of Parkin, which will release the auto-inhibitory structure of Parkin to activate its E3 ligase activity. However, how and when this Parkin conformational change occurs, especially in cells, remained unclear. In this paper, the authors made several Parkin mutants that disrupt intra-domain interactions and observed Parkin activation/recruitment/conformational change in cells by a FRET reporter assay they developed. Although the data they provided are clear and move the mitophagy research forward, the authors should answer the following major and minor comments before acceptance.

Major Comments

1) The conclusion for the deficiency of pUb binding coupled to W403A or F146A mutations is not clear. H302A/W403A or H302A/F146A delayed Parkin translocation, but they could not efficiently ubiquitinate Mfn2. On the other hand, they induced mitophagy as efficiently as the WT Parkin did. If the H302A mutation abolishes the interaction with phos-Ub built on the damaged mitochondria, Parkin (H302A) must not be recruited to the damaged mitochondria because phos-Ub acts as a

receptor for Parkin. One of the reasons that brought the vague conclusion is that H302A mutation does not completely abolish the interaction with pUb. Wauter et al (2015) Nature and Yamano et al (2015) J Biol Chem showed that impairment of the interaction with Ub I44 hydrophobic patch by introducing A320R mutation also completely disrupt the Parkin translocation. Therefore, Parkin mutant (single A320R or double H302A/A320R) may bring the clearer results. The authors should perform the experiments using at least one of the aforementioned mutants.

2) The authors concluded that the main role of the Ubl phosphorylation is to act as a lever, which can kick out the REP:RING1 and/or RING0:RING2 auto-inhibitory interfaces. However, the data the authors provided is not sufficient. To strength the conclusion, the authors may test the following experiments.

Compare the E2 enzyme accessibility among S65A, S65A/F146A, S65A/W403A, del-Ubl, del-Ubl/F146A, del-Ubl/W403A mutants.

Compare the C431 accessibility among S65A, S65A/F146A, and S65A/W403A del-Ubl, del-Ubl/F146A, del-Ubl/W403A mutants by C431 conjugation of ubiquitin vinyl-sulfone or other appropriate experiments.

3) FRET assay involves the most serious concern. Parkin should change the conformation upon mitophagy. Even for W403A and F146A mutants, Parkin needs Ubl phosphorylation and binding to phos-Ub for fully activation and/or the recruitment (neither W304A nor F146A mutation converts Parkin to constitutive-active form under normal condition). This means that, only under mitophagy-stimulation condition such as CCCP treatment, the different FRET from the basal state should be observed near the damaged mitochondria. As far as I understand, the authors' FRET assay was done only in Parkin basal state without mitophagy stimulation. Meanwhile, in Supplementary Figure 5c, FRET signal seems to change following CCCP treatment in FRET-81 and FRET-129. The authors should show FRET results +/- CCCP treatment in Fig. 8a-c. If such experiments are difficult following CCCP treatment because CFP/YFP moiety of FRET reporter is ubiquitylated as pseudo-substrate, FRET reporter with E3-deficient Parkin mutant (e.g., C431S) may be useful that can be targeted to depolarized mitochondria when co-expressed with wild-type Parkin [Zheng and Hunter (2013) Cell Research].

Minor Comments

1) Page 3: Add Yamano and Richard (2013) Autophagy to references for PINK1 degradation by the proteasome.

2) Page 3: Add Shiba-Fukushima et al (2012) Sci Rep. for Parkin Ubl phosphorylation and Kane et al (2014) J Cell Biol for ubiquitin phosphorylation.

3) Page 6: Add Yamano et al (2015) JBC for identification of pUb binding sites in Parkin.

4) Why did the authors use two different Ubl-deleted mutants, one is deletion of 1-76 for microscopy of Parkin recruitment and the other is deletion of 1-140 for in organelle ubiquitination assay? The authors should explain the reason in the main text.

5) Fig. 6C: The Parkin mutants on the x-axis should be H302A, but not R305A.

Point by Point Response to each reviewer's comments are highlighted in bold

Reviewer #1 (Remarks to the Author):

The Manuscript by M.Y. Tang et al. entitled `Structure-guided mutagenesis reveals a hierarchical mechanism of parkin activation` describes the activation mechanism of the protein Parkin which has been described as a key regulator of Parkinson's disease. A series of point mutations were selected based on a mentioned protein crystal structure and are further investigated in a series of assays for their effect on Parkin's ability to undergo mitochondrial translocation, ubiquitination, and supposed conformational changes investigated by FRET.

Overall, the manuscript has good potential. However, in its current state the manuscript suffers from some weaknesses, particular in the FRET section, that make it difficult to follow the author's conclusions.

Mitochondrial translocation assay:

What is the cut-off used by the authors to judge a cell as positive for the appearance of punctate GFP-parkin? If one looks at figure 1 panel b and compares the 60min picture for wt or N273K my rough estimate would be that wt is about 20-30% signal of N273K. However, if one compares this to the quantified time response data in panel c, the difference at time point 60min is roughly 25% between wt (75-80%) and N273K (100%). Is intensity considered at all, or would any weak spot be taken as positive punctate pattern? This discrepancy needs to be clarified and the assay criteria need to be better defined, since the assay is used throughout the manuscript and its results represent a major part of figures 1-4.

Response to Reviewer #1: We thank the reviewer for pointing out this apparent discrepancy. For the mitochondrial translocation assays, we consider the appearance of spots within the cells as positive puncta pattern. Intensity of the puncta was not taken into consideration. Images of Parkin recruitment shown in this manuscript are representative images of more than 350 cells imaged for each condition. Images were analyzed in a blind fashion using a methodology similar to what has been reported elsewhere (Narendra et al. 2008, Lazarou et al. 2012, and Shiba-Fukushima et al. 2014). We agree that the weaker intensity of the GFP-parkin puncta for WT (Figure 1b) may have caused some confusion, and have now updated Figure 1b that is more representative of the majority of cells imaged. We have also revised the methods and have added the following sentence: "The fluorescence intensity of the GFP-positive puncta was not taken into consideration for Parkin recruitment analysis".

FRET-probe: This section suffers from quite a number of weaknesses. First: Probe names. On page 11 the authors state: `The recruitment kinetics of Parkin with YFP inserted at positions 81 (after Ubl), 129 (Ubl-RINGO linker) and 380 (IBR-REP linker) were similar to the kinetics of N-terminally tagged Parkin. In contrast, insertion at positions 140 (before RINGO) delayed recruitment and insertion of YFP at positions 356 (within loop in IBR) and 465 (C-terminus) strongly impaired recruitment (Supplementary Fig. 5bc), and thus these constructs were not analyzed further. 465 (C-terminus) is irritating since the probe can be found as 466 (c-terminus) in table b supplementary figure 5. Please match nomenclature...

The text has been corrected and the nomenclature is consistent with Table B (new Suppl. Figure 10).

Position 140 shows delayed recruitment and 356 and 465 show strongly impaired recruitment. However, in table b supplementary figure 5 the recruitment after 1h CCCP is less for 140, much less for 356 and again less for 466. What are the author's criteria to convert delayed recruitment and strongly impaired recruitment to less or much less?

The reviewer makes a good point that we did not adequately outline the criteria for “delayed recruitment”. Based on the comments, we have now performed mitochondrial translocation assays for our FRET reporters and have now included this quantification in Suppl. Figure 11. The FRET-81, FRET-129, and FRET-380 reporters had greater than 70% Parkin recruitment after 90 mins CCCP treatment, whereas FRET-140, FRET-355, and FRET-466 had recruitment in less than 40% of cells. Using this quantification, it is clear that our FRET constructs fall into two groups, with FRET-81, FRET-129, and FRET-380 behaving similarly to YFP-Parkin WT, whereas FRET-140, FRET-355 and FRET-466 were delayed and therefore no longer considered.

Second: The description of the probe design is extremely vague and it is currently impossible for a skilled person to reproduce any of their findings. On page 16 in the methods section the authors state: The sequence of Venus from the Venus-N1 construct was inserted at various sites within Parkin by PCR and fragment exchange (DNA Express Inc.). The resulting construct was amplified by PCR using primers containing BglIII and HindIII. This is the shortest and most insufficient description of a FRET probe I have seen in a long time! The authors need to state the use of linkers or should state that no linkers were used. If no linker were used, how does the distance in N-term and C-term of Venus influence the parkin protein? The insertion of Venus will split parkin in two parts and eventually insert a distance of up to 2.2 nm between both fragments (2.2 nm by simply measuring the distance in GFP pdb code 1emb, Glu6 to Ile229 given as 22.38 Å). This needs to be taken into account and discussed!

We now provide the complete sequence of our FRET reporter constructs (Suppl. Figure 14), and we have updated our method section to include more details on how the constructs were made. In short, no linkers were used. While we agree that the insertion of Venus splits Parkin in two parts separated by roughly 2.0-2.5 nm, these insertion points were chosen to be loops and linkers within Parkin where weak or no electron density was observed (except FRET-355 and FRET-466). For instance, PDB 4K95 (full-length rat Parkin) had no electron density between residues 73 and 140 (between UBL and RING0), and all Parkin structures (rat, human, louse) show weak or no electron density between residues 380 and 390 (between IBR and REP). The inserted Venus is therefore likely to be accommodated by the flexibility of the surrounding residues, without affecting much the intramolecular interactions that regulate Parkin's activity. Our functional data indeed suggest that for the reporter constructs FRET-81, FRET-129 and FRET-380, the insertion neither affect its ability to autoubiquitinate and respond to PINK1 phosphorylation, nor its function in mitophagy (Suppl. Figure 11 & 12). We have updated the results and discussion sections to include these points.

Figure 7 FRET-129: The structural illustration of YFP is strongly misleading since it is connected to parkin via a single amino acid link. If it is inserted into parkin, it has to have two connections to parkin.

We agree that our original figure was misleading. Part of the confusion stemmed from the fact that there is no atomic model for the linker between the Ubl and RING0 domains (residues 81-140). We have therefore drawn lines between the corresponding parts of the structures that are tethered in the reporter constructs (new Figure 7). We hope the new figure helps dispel any mis-interpretation.

The distance interpretation in all FRET Probes has to be revised by the authors with great care. An inserted Venus is restrained in its degrees of freedom and hence, unlikely to freely rotate as a fluorescent protein could do if it was tethered at the N- or C-term of a protein. Therefore, any distance interpretation is a wild guess since the assumption of $K=2/3$ as value for the Förster equation does not hold true. It could be anything between 0 and 4. Without anisotropy measurements, it is impossible to draw any conclusion with respect to distances! Therefore, the interpretation of the results needs to be revised with greater care!

We agree with the reviewer that the inserted Venus may be restrained. However, as indicated above, Venus was inserted in flexible linkers, and therefore may be able to rotate. Nonetheless, we agree that it would be hazardous to attempt any kind of quantitative distance assessment based on our FRET data. Our original goal was to perform qualitative analysis and empirical FRET measurement in Parkin mutants that correlate with specific functional outcomes. We don't know how the Cerulean and Venus domains will orient in the reporter constructs, nor how they might rearrange in the activating mutants. Still, we observe FRET intensities and changes that are consistent with our expectations. We predicted FRET-380 to give the strongest basal FRET values, and this is indeed what we observe. We also expected FRET-380 to show a reduction in FRET in the N273K and W403A mutants. However, restricted motions and anisotropic effects may explain why FRET-81 shows an increase in FRET in the N273K mutant. To dispel any impression that we are performing precise distance measurements, we have therefore removed the distance ranges in the new Figure 7. We have also amended the results and discussion sections to account for these different effects.

Minor Points:

Page 5:

`The crystal structure of Parkin revealed a number of inter-domain contacts that maintain auto-inhibition of its E3 ubiquitin ligase activity (Fig. 1a). Please provide the reference for the mentioned crystal structure, since it is very irritating for none specialists to go to figure 1 and discover a cartoon rather than a structure as expected after this sentence...

We have modified the sentence as following: “The crystal structure of Parkin (PDB code 4K95) revealed a number of inter-domain contacts that maintain auto-inhibition of its E3 ubiquitin ligase activity (Fig. 1a).” In the legend of Fig. 1a we also state explicitly that we are just showing a cartoon representation of Parkin's structure and activation model, and not the crystal structure itself.

Figure 3. The labeling of the individual figure panels is from a-d and does not match the labeling in the figure legend a-b, e-f. Obviously or likely, the figure has been modified from a previous version. Please correct labeling in the figure legend to a-d.

We have corrected the Figure legend in Figure 3.

Last but not least: I would be interested to know if any of the described mutations occur in patients suffering from Parkinson's disease. Is there anything known?

We have looked for polymorphism from Parkinson's disease patients from a dataset consisting of 520 Parkinson's disease patients and 690 controls and did not find anyone that carried the N273K, W403A or F146A variants. We also searched in the Exome Aggregation Consortium (ExAC) database that includes data from more than 60 000 unrelated individuals (no necessarily from Parkinson's disease studies) and did not find these mutations there either. This likely means that these variants are very rare or do not exist. We emphasize in the text that these mutations were used as a proof in principle to show that Parkin can be activated using single point mutations that disrupt its autoinhibited conformation, and no way do we imply that these mutations are risk factors or protective factors for Parkinson's disease.

Reviewer #2 (Remarks to the Author):

There is a pressing need therapeutically and biologically to understand how the Parkin E3 ligase that is mutated in Parkinson's disease is activated. The Fon group previously solved the crystal structure of Parkin that revealed Parkin to be in a complex auto inhibited conformation at several interfaces. Together with other groups they have demonstrated that phosphorylation of Parkin at its Ubl domain and of ubiquitin together lead to rearrangement of the Parkin structure into an active conformation with concomitant activation of Parkin E3 ligase activity. In this paper they address mechanistically the step-wise order by which each of these autoinhibitory interfaces are released to lead to activation. Overall they provide evidence that phospho-ubiquitin binding to Parkin is the initial step in activation and that Parkin phosphorylation is not simply leading to a release of the Ubl domain but plays an active role in causing release of the REP element that binds RING1 and the RING0/RING2 auto inhibition towards activation. The analysis undertaken is elegant and the results extremely clear.

Whilst several of their conclusions have already been made in previous studies published last year and this year by other groups, their approach is distinct due to the comprehensive nature of the analysis undertaken to study all the auto inhibitory interfaces together and using both in vitro and cellular assays of Parkin recruitment, activity and mitophagy.

Overall I have only a few technical suggestions to strengthen the conclusion of the paper:

1. Whilst they show that the Ubl/RING1 disruptor mutant, N273K, is more active than WT in Parkin recruitment and activity (Fig1b-d), this mutant is totally unable to rescue defects in Parkin Ser65

phosphorylation. This is not completely expected given the increased activity and could there be a specific interaction here between the N273K and S65A residues that blocks activity. It would be important to test different Ubl domain mutants that have previously shown to affect the Ubl/RING1 interface e.g. the R33Q or A46P to confirm this result.

We thank the reviewer for suggesting additional experiments to test different Ubl domain mutations. Although mutations within the Ubl (A42P, A46P, R33Q) are able to release Parkin's auto-inhibitory interaction between the Ubl and RING1 and increase E3 ligase activity in vitro (Chaugule et al. 2011), the expression of R42P, R46P, or I44A in cells were shown to delay Parkin recruitment to the mitochondria (Narendra et al. 2010). This is in contrast to the N273K mutation that we show increases Parkin recruitment. Interestingly, the I44A mutation reduces its ability to be phosphorylated by PINK1 (Wauer et al. 2015), suggesting that mutations in the Ubl disrupt its interaction with PINK1. Given that these mutations impair rather than activate Parkin, we therefore reasoned that they would not be able to be rescued. However, we did explore Δ Ubl, which on its own is impaired, and found it cannot be rescued by N273K, which is consistent with our S65A rescue experiments.

2. The effects of the activating mutations on mitophagy at 4h show no effect with the N273K mutant despite this also being more active than WT. It would be good to see a time course of CCCP-induced mitophagy to compare the mutants with WT more comprehensively.

We agree that a more comprehensive time-course of CCCP-induced mitophagy is a good way to determine if there are any differences between WT and N273K at earlier time points (before 4 hours). We therefore analyzed mitophagy using our mito-Keima FACS assay and found that the N273K mutant had no effect at shorter CCCP time points (new Suppl. Figure 7).

3. The translocation assay to check FRET constructs is not sufficient to assess impact on function. To verify that the insertion of YFP into residues 81, 129 and 380 is silent, it would be important to look at other downstream readouts of Parkin activity in cells upon CCCP and also express them in E.coli to check in vitro activity.

To further assess the function of the FRET-reporters, we tested these reported constructs in three different assays: mitochondrial recruitment, mitophagy (mito-Keima), and autoubiquitination. We first find that the recruitment kinetics of FRET-81 and FRET-129 and FRET-380 are similar to the kinetics of single YFP-labeled Parkin (Suppl. Figure 11). FRET-140, FRET-355 and FRET-466 were impaired and were therefore not further analyzed. We then found that the same three reporter constructs that were effective in mitochondrial recruitment were also capable of inducing mitophagy (Suppl. Figure 12a-b). In addition, we purified recombinant FRET-reporter proteins from *E.coli*, as suggested by the reviewer, and tested their E3 ligase activity with an *in vitro* auto-ubiquitination assays in the presence of DpPINK1 (Suppl. Figure 12c). The data shows clearly that these reported constructs can auto-ubiquitinate and can be activated by PINK1 phosphorylation. We have revised the text on pp.11-12 to highlight these important controls.

Minor points:

1. On page 5, results section, the times stated for WT parkin recruitment of 30mins post CCCP does not reflect the data shown in Fig1 and supp Fig 1. Also true for N237K lead time mutant.

The time stated for WT parkin recruitment of 30 min is the time in which GFP-puncta first begin to appear. However, not all of the cells show Parkin recruitment at 30 mins which is evident from the quantification graph of Parkin recruitment shown in Figure 1c. Thus, the data shown in Figure 1b is for illustration purpose and reflects Parkin recruitment observed in the majority of cells at the indicated time points.

2. Bottom of page 7 - should read Fig 2e and not 3e

3. The labelling of Fig 6c x-axis should be switched from R305A to H302A

These errors have been corrected.

Reviewer #3 (Remarks to the Author):

Parkin and PINK1 are the gene products of familial Parkinson's disease, and are essential for the clearance of damaged mitochondria. PINK1 accumulated on the damaged mitochondria phosphorylates both ubiquitin and the Ubl domain of Parkin, which will release the auto-inhibitory structure of Parkin to activate its E3 ligase activity. However, how and when this Parkin conformational change occurs, especially in cells, remained unclear. In this paper, the authors made several Parkin mutants that disrupt intra-domain interactions and observed Parkin activation/recruitment/conformational change in cells by a FRET reporter assay they developed. Although the data they provided are clear and move the mitophagy research forward, the authors should answer the following major and minor comments before acceptance.

Major Comments

1) The conclusion for the deficiency of pUb binding coupled to W403A or F146A mutations is not clear. H302A/W403A or H302A/F146A delayed Parkin translocation, but they could not efficiently ubiquitinate Mfn2. On the other hand, they induced mitophagy as efficiently as the WT Parkin did. If the H302A mutation abolishes the interaction with phos-Ub built on the damaged mitochondria, Parkin (H302A) must not be recruited to the damaged mitochondria because phos-Ub acts as a receptor for Parkin. One of the reasons that brought the vague conclusion is that H302A mutation does not completely abolish the interaction with pUb. Wauter et al (2015) Nature and Yamano et al (2015) J Biol Chem showed that impairment of the interaction with Ub I44 hydrophobic patch by introducing A320R mutation also completely disrupt the Parkin translocation. Therefore, Parkin mutant (single A320R or double H302A/A320R) may bring the clearer results. The authors should perform the experiments using at least one of the aforementioned mutants.

We thank the reviewer for suggesting experiments with other pUb-binding Parkin mutants. We have therefore tested whether our activating mutations can rescue the A320R variant in our Parkin recruitment assays (Suppl. Figure 2). Cells expressing A320R or A320R/N273K mutants were unable to recruit to depolarized mitochondria, whereas the A320R/W403A and A320R/F146A mutants showed a considerable delay compared to WT (this delay is greater than what we observed with H302A/W403A or H302A/F146A). However, both the W403A and F146A double mutants, but not the N273K, were able to rescue A320R in the mt-Keima mitophagy assay (Suppl. Figure 8), which is similar to the results obtained with the H302A mutant (Figure 6). We argue that the fast recruitment kinetics is not essential for maximal mitophagy, and the Parkin recruitment seen at later time points (120 mins) is able to overcome the reduced pUb-binding affinity of the H302A or A320R Parkin mutants in our mt-Keima mitophagy assay. These experiments are consistent with our previous findings that disruption of either the REP:RING1 or RING0:RING2 interface can rescue mitophagy deficits in Parkin mutants impaired in pUb binding (Figure 2). We have incorporated these findings in the Result section titled “Phospho-ubiquitin binding precedes Parkin phosphorylation and activation”.

2) The authors concluded that the main role of the Ubl phosphorylation is to act as a lever, which can kick out the REP:RING1 and/or RING0:RING2 auto-inhibitory interfaces. However, the data the authors provided is not sufficient. To strength the conclusion, the authors may test the following experiments. Compare the E2 enzyme accessibility among S65A, S65A/F146A, S65A/W403A, del-Ubl, del-Ubl/F146A, del-Ubl/W403A mutants. Compare the C431 accessibility among S65A, S65A/F146A, and S65A/W403A del-Ubl, del-Ubl/F146A, del-Ubl/W403A mutants by C431 conjugation of ubiquitin vinyl-sulfone or other appropriate experiments.

We have performed an Ub_{CH7}~Ub discharging assays to indirectly assess E2 enzyme accessibility (new Suppl. Figure 5a,b). We have previously shown that the W403A mutant, which binds more strongly to Ub_{CH7} than WT, discharges Ub_{CH7}~Ub faster (Trempe et al. 2013). Our new results show that phospho-Parkin indeed discharges faster than WT, as expected. The W403A and S65A/W403A both discharged Ub_{CH7}~Ub faster than WT, but not as fast as phospho-WT. Moreover, the Δ Ubl-W403A also discharged Ub_{CH7}~Ub faster than Δ Ubl. On the other hand, the N273K and S65A/N273K mutant did not significantly enhance discharging, which is consistent with the inability of the N273K mutant to rescue S65A in organello, in Parkin recruitment and mitophagy. We excluded the F146A and S65A/F146A mutants from our analysis, since these mutants had poor solubility, presumably because the F146A mutation exposes hydrophobic surfaces at the surfaces of both RING0 and RING2. Indeed, we observed precipitate formation for these mutants following incubation above room temperature at the concentration used for the discharging assays (1 μ M). These F146A mutants give much lower expression yields in *E. coli*, as opposed to the W403A mutation, which is similar to WT. The F146A mutants could be tested in organello because the concentration used (100 nM) was lower than in the discharging assays. The lower solubility, as well as different FRET values observed for the F146A mutants and their interpretations with regards to the conformation of these mutants are now discussed on page 15.

Next, we assessed exposure of Cys431 using HA-Ub-vinyl sulfone (new Suppl. Figure 5c,d). As reported by others previously, we observe that phospho-Parkin reacts more strongly with Ub-VS. The W403A mutant also increased reactivity towards Ub-VS in the context of the WT, S65A and Δ Ubl proteins. Intriguingly, the S65A and N273K mutations mildly increased reactivity towards Ub-VS. This suggests that E2~Ub discharging is a better predictor of Parkin's activity in mitophagy than C431 reactivity. As these experiments are carried out in the absence of E2 or Ub~E2, it also suggests that while S65A and N273K appear to change parkin conformation to make C431 more accessible to Ub-VS, this is unlikely to represent a conformation parkin naturally adopts when undergoing E2-dependent Ub transfer.

These new results (discharging and Ub-VS) are now described in the main results section on page 9, which we have re-worded to integrate it with the *in organello* and cellular assays in the same section.

3) FRET assay involves the most serious concern. Parkin should change the conformation upon mitophagy. Even for W403A and F146A mutants, Parkin needs Ubl phosphorylation and binding to phospho-Ub for fully activation and/or the recruitment (neither W304A nor F146A mutation converts Parkin to constitutive-active form under normal condition). This means that, only under mitophagy-stimulation condition such as CCCP treatment, the different FRET from the basal state should be observed near the damaged mitochondria. As far as I understand, the authors' FRET assay was done only in Parkin basal state without mitophagy stimulation. Meanwhile, in Supplementary Figure 5c, FRET signal seems to change following CCCP treatment in FRET-81 and FRET-129. The authors should show FRET results +/- CCCP treatment in Fig. 8a-c. If such experiments are difficult following CCCP treatment because CFP/YFP moiety of FRET reporter is ubiquitylated as pseudo-substrate, FRET reporter with E3-deficient Parkin mutant (e.g., C431S) may be useful that can be targeted to depolarized mitochondria when co-expressed with wild-type Parkin [Zheng and Hunter (2013) Cell Research].

The reviewer is correct to point out that the FRET analysis was done without mitophagy stimulation. Upon CCCP addition, a change in FRET efficiency is expected to be observed near the site of damaged mitochondria as Parkin is converted into an active form. When we treated our FRET-reporters with CCCP to assess recruitment kinetics, we found that there are indeed changes in FRET efficiency. However, we found that all of the FRET-constructs (except FRET-380) resulted in an increase in FRET efficiency in regions of interest associated with the mitochondria (see Figure below). Furthermore, we observed an increase FRET-efficiency with CFP and YFP on separate Parkin molecules used as controls. We suspect that the effect results from molecular crowding, since we could not saturate the FRET by varying the donor to acceptor ratio (results not shown). This effect is consistent with a previous report that observed Parkin oligomerization at mitochondria by fluorescence correlation microscopy (Lazarou et al. 2013). Because the changes in FRET we observe result from both intra- and intermolecular interactions, it will be difficult to estimate the contribution of intramolecular conformation changes that result from the activation of Parkin using CCCP. For these reasons, our FRET-reporters will not be informative to follow conformational changes associated with CCCP induced mitophagy. We have therefore removed the original panel (Supplementary Figure 5c) showing FRET images of the FRET-reporters treated with CCCP from the revised manuscript. However, we believe that our FRET-reporters will be an important tool for investigating Parkin activation in

response to chemical inducers of cellular stress or activators of survival pathways which does not cause massive damage to the mitochondria. For the interest of this manuscript, we would like to emphasize that we are using fluorescent probes to highlight the conformational changes associated with point mutations designed to de-repress Parkin's autoinhibited conformation.

FRET construct	DMSO	CCCP (diffuse)	CCCP (punctate)
CFP-pk + YFP-pk	0.01	0.02	0.27
FRET-81	0.24	0.29	0.44
FRET-380	0.45	0.42	0.44

Minor Comments

1) Page 3: Add Yamano and Richard (2013) Autophagy to references for PINK1 degradation by the proteasome.

2) Page 3: Add Shiba-Fukushima et al (2012) Sci Rep. for Parkin Ubl phosphorylation and Kane et al (2014) J Cell Biol for ubiquitin phosphorylation.

3) Page 6: Add Yamano et al (2015) JBC for identification of pUb binding sites in Parkin.

These references have been added to the text.

4) Why did the authors use two different Ubl-deleted mutants, one is deletion of 1-76 for microscopy of Parkin recruitment and the other is deletion of 1-140 for in organelle ubiquitination assay? The authors should explain the reason in the main text.

Recombinant Δ 1-76 Parkin had a tendency to degrade in vitro, whereas Δ 1-140 (R0RBR) was more stable. However, we had previously tested Δ 1-76 and Δ 1-76 -W403A in organello, and they both behaved similarly to Δ 1-140 (new Suppl. Fig. 4). We updated the text accordingly on page 9.

5) Fig. 6C: The Parkin mutants on the x-axis should be H302A, but not R305A.

The figure legends for Fig. 6C have been fixed.

Reviewers' Comments:

Reviewer #1 (Remarks to the Author)

The authors have done a great amount of work to address the criticism that was raised. They have answered all of my questions and more. I have no more comments, except that I am glad to see that the clarity of manuscript has improved significantly.

Reviewer #2 (Remarks to the Author)

The paper is substantially improved following the revisions.

Reviewer #3 (Remarks to the Author)

Ms. No. NCOMMS-16-13984A

Title: Structure-guided mutagenesis reveals a hierarchical mechanism of Parkin activation

Accept

In the revised paper, the authors adequately addressed to our comments, and this revised manuscript is now deemed suitable for acceptance in Nature Communications, although the authors should address the following two minor comments.

Minor comment 1: Supple Fig12 C, Is TcPINK1 in the figure typo for DpPINK1?

Minor comment 2: In Introduction, the authors say that "we show that binding to pre-existing pUb on mitochondria is essential for Parkin phosphorylation, and indeed the H302A mutant cannot be rescued by any of the activating mutations". However, the authors show that W403A and F146A mutants could rescue the H302A mutations in the mt-Keima mitophagy assay (Fig. 6), and so this sentence might confuse readers. The authors should describe the sentence more carefully such as "indeed deficits in Parkin recruitment and ubiquitination by the H302A mutant cannot be rescued by any of the activating mutations".

Anyway, I think this paper will be a solid springboard for future discussion regarding Parkin activation mechanism, and will be cited many times as other reliable papers published by J-F and Ted.

Point by Point Response to each reviewer's comments are highlighted in bold

REVIEWERS' COMMENTS:

Reviewer #1 (Remarks to the Author):

The authors have done a great amount of work to address the criticism that was raised. They have answered all of my questions and more. I have no more comments, except that I am glad to see that the clarity of manuscript has improved significantly.

Reviewer #2 (Remarks to the Author):

The paper is substantially improved following the revisions.

We thank Reviewer #1 and #2 for their comments and suggestions.

Reviewer #3 (Remarks to the Author):

Ms. No. NCOMMS-16-13984A

Title: Structure-guided mutagenesis reveals a hierarchical mechanism of Parkin activation

Accept

In the revised paper, the authors adequately addressed to our comments, and this revised manuscript is now deemed suitable for acceptance in Nature Communications, although the authors should address the following two minor comments.

Minor comment 1: Supple Fig12 C, Is TcPINK1 in the figure typo for DpPINK1?

The original manuscript incorrectly stated the use of recombinant TcPINK1. In fact, recombinant DpPINK1 was used for all of the *in vitro* ubiquitination assays. We apologize for this error.

Minor comment 2: In Introduction, the authors say that “we show that binding to pre-existing pUb on mitochondria is essential for Parkin phosphorylation, and indeed the H302A mutant cannot be rescued by any of the activating mutations”. However, the authors show that W403A and F146A mutants could rescue the H302A mutations in the mt-Keima mitophagy assay (Fig. 6), and so this sentence might

confuse readers. The authors should describe the sentence more carefully such as “indeed deficits in Parkin recruitment and ubiquitination by the H302A mutant cannot be rescued by any of the activating mutations”.

As suggested, we have revised the sentence in the introduction on p5 as follows:

On the other hand, we show that binding to pre-existing pUb on mitochondria is essential for Parkin phosphorylation, and indeed deficits in Parkin recruitment and ubiquitination of outer membrane substrates by the H302A mutant cannot be rescued by any of the activating mutations.

Anyway, I think this paper will be a solid springboard for future discussion regarding Parkin activation mechanism, and will be cited many times as other reliable papers published by J-F and Ted.

We would like to thank Reviewer #3 for their comments and suggestions.